# Distinctive Features and Fabrication Routes of Metallic-Glass Systems Designed for Different Engineering Applications: A Review

**Akib Jabed** [1], **M. Nabil Bhuiyan** [2], **Waseem Haider** [3,4] **and Ishraq Shabib** [3,4,*]

1 Department of Mechanical Engineering, The University of Texas at Dallas, Richardson, TX 75080, USA; akib.jabed@utdallas.edu
2 Department of Materials Science and Engineering, University of Connecticut, Storrs, CT 06269, USA; m_nabil.bhuiyan@uconn.edu
3 School of Engineering & Technology, Central Michigan University, Mt. Pleasant, MI 48859, USA; haide1w@cmich.edu
4 The Science of Advanced Materials, Central Michigan University, Mt. Pleasant, MI 48859, USA
* Correspondence: shabi1i@cmich.edu

**Abstract:** Materials with a disordered atomic structure, often termed glassy materials, are the focus of extensive research due to the possibility of achieving remarkable mechanical, electrochemical, and magnetic properties compared to crystalline materials. The glassy materials are observed to have an improved elastic modulus combined with a higher strength and hardness. Moreover, better corrosion resistance in different mediums is also observed for glassy solids, which is difficult to attain using conventional crystalline materials. As a result, the potential applications of metallic-glass systems are continually increasing. Amorphous materials are usually divided into two categories based upon their size. Materials with a thickness and diameter larger than the millimeter (mm) scale are termed as bulk metallic glass (BMG). However, the brittle nature of the bulk-sized samples restricts the size of metallic-glass systems to the micron (μm) or nanometer (nm) range. Metallic glasses with a specimen size in the scale of either μm or nm are defined as thin-film metallic glass (TFMG). In this review, BMGs and TFMGs are termed as metallic glass or MG. A large number of multi-component MGs and their compositional libraries reported by different research groups are summarized in this review. The formation of a multicomponent metallic glass depends on the constituent elements and the fabrication methods. To date, different unique fabrication routes have been adopted to fabricate BMG and TFMGs systems. An overview of the formation principles and fabrication methods as well as advantages and limitations of conventional MG fabrication techniques is also presented. Furthermore, an in-depth analysis of MG inherent properties, such as glass forming ability, and structural, mechanical, thermal, magnetic, and electrochemical properties, and a survey of their potential applications are also described.

**Keywords:** metallic glass; glass forming ability; mechanical properties; corrosion

## 1. Introduction

Metallic glasses (MG) are disordered non-equilibrium solids that, unlike crystalline alloys, lack long-range order. Metallic-glass systems are multicomponent with substantial differences in atomic sizes selected to suppress the crystallization of the liquid melt; extremely high cooling rates are used for vitrification [1]. Although metallic glasses possess high strength and a disordered structure like conventional glasses, high glass-transition temperatures are required for MGs [1]. Early work on metallic-glass systems started in the 1960s and 1970s [2–4]. The first disordered MG was a gold–silicon system developed by Jun et al. [2] and was fabricated via the melt cooling technique. Later on, Pd-Si-based metallic-glass systems (binary, ternary, and quaternary) with thicknesses of more than 1

mm were produced via the splat quenching technique under different cooling rates at room temperature by Chen et al. [3]. In 2000, metallic–glass systems (e.g., Fe-, Co- and Ni-based) were produced with thicknesses in the micrometer scale using cooling rates of $\sim 10^5$ K/s [5].

Early metallic-glass systems were usually fabricated via the rapid quenching method to vitrify the liquid melt, thus requiring an extreme rate of cooling and introducing a size limitation on the metallic-glass systems [6]. However, $Pd_{40}Ni_{40}P_{20}$ [7,8] and Pd-Cu-Si [4] glass systems are reported to have been fabricated with thicknesses of 10 mm using low cooling rates ($\sim 10$ K/s). These systems were found to have better glass-forming ability, but the high cost of palladium (Pd) hindered the practical implementation. The Zr-based (Zr-Ni-Al) metallic-glass systems reported by Inoue et al. [9] are considered to be the "earliest significant advancement" in glassy solids. Zr-based systems are observed to display excellent mechanical strength with a wide supercooled region, which can be useful for micro- or nano-fabrication [9,10]. The potential of Zr-based systems was further explored by Peker et al. [11], who included beryllium (Be) in Zr-Cu-Ni-Al. It is noteworthy that Be is the smallest atom in this metallic-glass system and its inclusion stabilizes the liquid and glassy phase [11]. This Be-containing amorphous alloy is a good illustration of the requirement for multiple species to form stable metallic glasses, chosen with a significant atomic size ratio.

The formation of metallic glasses also depends on the fabrication route. Solidification of the liquid melt through quenching is found to be helpful to maintain the glassy state; however, the very high cooling rate suggests an alternative technique. Conventional copper mold casting is found to be useful in this regard [6,12]. Lower quench rates of the order of 0.067–0.133 K/s have been reported for the fabrication of several metallic-glass systems [13], including $Zr_{55}Cu_{30}Ni_5Al_{10}$ [14], $Pd_{40}Cu_{30}Ni_{10}P_{20}$ [15], $Ni_{50}Pd_{30}P_{20}$ [16], $Mg_{59.5}Cu_{22.9}Ag_{6.6}Gd_{11}$ [17], $Cu_{36}Zr_{48}Ag_8Al_8$ [18], $Fe_{48}Cr_{15}Mo_{14}Er_2C_{15}B_6$ [19], and $Ti_{40}Zr_{10}Cu_{32}Pd_{14}Sn_4$ [20]. These systems are reported to have larger diameters ($\sim 1$–2 cm), which are useful in different engineering applications [12]. Nevertheless, the quench rates for the production of the amorphous solids still require optimization, and vapor-quenching techniques are found useful in this regard. Thin metallic-glass systems are produced using vapor-quenching techniques as reported in ref. [21–24]. The metallic glasses formed through the vapor quenching method are less brittle. Furthermore, the composition of the multicomponent systems also plays a vital part in glass formation. Deep eutectic points are found to be useful for obtaining amorphous structures, and choosing a composition near the eutectic point facilitates the formation of a metallic glass [25].

Metallic-glass systems are found to exhibit outstanding mechanical [26–28], electrochemical [29,30], and magnetic properties [31,32] due to the absence of dislocations and grain boundaries. In addition, using the glass-transition temperature of the metallic glass systems, they can be formed into unique structures by simple molding or drawing. [33,34]. Such outstanding features of the metallic-glass systems enable them to be used in biomedical systems [35,36], micro-electro-mechanical systems (MEMS) [37,38], nano-electro-mechanical systems [39,40], catalysis [41–43], and structural [44,45] applications. In this review, different fabrication techniques of the metallic-glass systems are described in combination with their unique properties and potential applications.

## 2. Glass Forming Ability

Glass forming ability (GFA) and its relationship with the atomic structure is crucial to the properties of metallic glasses. While discussing GFA, the "Confusion Principle" proposed by Lindsay Greer should be mentioned [46]. According to Greer, glass formation is facilitated due to increasing "confusion" during crystallization. The principle is primarily focused on the crystal structure and the atomic size of the components in multispecies systems [46]. Thus, the use of an additional element in the composition, such as a small beryllium atom, can facilitate metallic-glass formation [46]. The term "confusion" is further defined by Perim et al. [47], who states that impeding the growth of critical-size nuclei during synthesis can lead to the formation of an amorphous-state structure. According

to this theory [47], the presence of multiple phases with different structures manifesting similar energies should form a metallic glass.

The empirical rules of metallic glass formation are further explained by Inoue et al. [5], who postulates three determining parameters for the possible formation of metallic glass: (i) the presence of multiple species, (ii) atomic size mismatch ($\geq$12%) among the species, and (iii) the negative heat of mixing. The extensive classification of metallic-glass systems based on these parameters has also been reported Takeuchi et al. [48]. Moreover, compositional analyses of metallic-glass systems reveals that an optimum composition is required to achieve the maximum supercooled region. On the other hand, only 32 elements have been used from the periodic table for the analysis of metallic glass formation and a significant number of these combinations are produced through trial and error by different research groups [49]. For example, a study carried out by Zeman et al. [50] assessed the optimum composition for Zr-Cu binary systems by varying the copper content from 18 to 88 at.%. They found that a Cu content within the range 30–65 at.% facilitates glass transition; at 55 at.% of Cu the maximum supercooled region is achieved.

Deviation from the glass-forming rules of Inoue is often observed as the method of fabrication also plays a vital role in the formation of metallic glass. The most commonly reported metallic-glass formation techniques are liquid-to-solid transition methods, which require a high range of critical cooling rates [51–53]. However, a high cooling rate hampers the production of large-size metallic-glass systems and may contribute to partial crystallinity in the material matrix [54]. For example, Calin et al. [54] fabricated $Ti_{75}Zr_{10}Si_{15}$ and $Ti_{60}Nb_{15}Zr_{10}Si_{15}$ systems via arc melting. The first system possessed a positive heat of mixing and a Ti-Zr atomic mismatch below 12%, and their results revealed a partially crystalline metallic glass structure. For the quaternary system, the Ti-Nb bond exhibited a positive heat of mixing, and the atomic mismatch also did not fulfil Inoue's rule; in that case as well, the synthesis led to the formation of a metallic glass with partial crystallinity. These issues can be solved with a higher rate of vapor quenching [13]. Metallic-glass systems produced via different techniques are listed in Table 1 and the systems are assessed with respect to Inoue's empirical rule.

**Table 1.** Metallic-glass systems and corresponding parameters associated with Inoue's empirical glass-forming rules [5,48]. The heat of mixing values was obtained from refs. [5,48] and the atomic size mismatch was calculated by using the atomic radius [54]. The MG fabrication technique is also listed.

| Metallic Glass System | Atomic Size Mismatch (%) ($r_{base} - r_O$)/$r_{base}$ | Heat of Mixing (kJ/mol) | Fabrication Route | Ref. |
|---|---|---|---|---|
| $Zr_{41.2}Ti_{13.8}Cu_{12.5}Ni_{10}Be_{22.5}$ | Zr:Ti = 9.6; Zr:Cu = 20 Zr:Ni = 22.3 Zr:Be = 29.6 | Zr-Ti = 0; Zr-Cu = −23 Zr-Ni = −49; Zr-Be = −43 Ti-Cu = −9; Ti-Ni = −35 Ti-Be = −30; Cu-Ni = 4 Cu-Be = 0; Ni-Be = −4 | Casting in copper molds | [11] |
| $Pt_{57.5}Cu_{14.7}Ni_{5.3}P_{22.5}$ | Pt:Cu = 7.8 Pt:Ni = 10.2 Pt:P = 23.5 | Pt-Cu = −12; Pt-Ni = −5 Pt-P = −34.5; Cu-Ni = 4 Cu-P = −17.5; Ni-P = −34.5 | Water quenching | [55] |
| Zr-Ti-Nb-Cu-Be | Zr:Ti = 9.6 Zr:Nb = 10.7 Zr:Cu = 20 Zr:Be = 29.6 | Zr-Ti = 0; Zr-Nb = 4 Zr-Cu = −23; Zr-Be = −43 Ti-Nb = 2; Ti-Cu = −9 Ti-Be = −30; Nb-Cu = 3 Nb-Be = −25; Cu-Be = 0 | Arc-melting and heated via induction | [56] |
| $Cu_{47.5}Zr_{47.5}Al_5$ | Cu:Zr = 20 Cu:Al = 10.7 Zr:Al = 10.6 | Cu-Zr = −23; Cu-Al = −1 Zr-Al = −44 | Arc-melting | [57] |

**Table 1.** *Cont.*

| | | | | |
|---|---|---|---|---|
| $Pd_{40}Ni_{40}P_{20}$ | Pd:Ni = 9.4<br>Pd:P = 23<br>Ni:P = 14.8 | Pd-Ni = 0; Pd-P = −36.5<br>Ni-P = −34.5 | Fluxing | [8] |
| $Zr_{39}Cu_{39}Ag_{22}$ | Zr:Cu = 20<br>Zr:Ag = 9.8 | Zr-Cu = −23; Zr-Ag = −20;<br>Cu-Ag = 2 | DC reactive magnetron sputtering | [58] |
| $Zr_{59}Ti_{22}Ag_{19}$ | Zr:Ti = 9.6<br>Zr:Ag = 9.8 | Zr-Ti = 0; Zr-Ag = −69; Ti-Ag = −2 | Magnetron sputtering | [59] |
| Zr-Ti-Fe | Zr:Ti = 9.6<br>Zr:Fe = 22.5<br>Ti:Fe = 8.8 | Zr-Ti = 0; Zr-Fe = −25;<br>Ti-Fe = −17 | Magnetron co-sputtering | [60] |
| Zr-Ni-Al-Si | Zr:Ni = 22.3<br>Zr:Al = 10.6<br>Zr:Si = 28 | Zr-Ni = −49; Zr-Al = −44;<br>Zr-Si = −84; Ni-Al = −22;<br>Ni-Si = −40; Al-Si = −19 | RF and DC reactive magnetron sputtering | [61] |
| Zr-Cu-Al-Ag | Zr:Cu = 20<br>Zr:Al = 10.6<br>Zr:Ag = 9.8 | Zr-Cu = −23; Zr-Al = −44<br>Zr-Ag = −69; Cu-Al = −1<br>Cu-Ag = 2; Al-Ag = −4 | DC magnetron sputtering | [62] |
| $Cu_{48}Zr_{42}Ti_4Al_6$ | Cu:Zr = 20<br>Cu:Ti = 14.3<br>Cu:Al = 10.7 | Cu-Zr = −23; Cu-Ti = −9<br>Cu-Al = −1; Zr-Ti = 0<br>Zr-Al = −44; Ti-Al = −30 | RF magnetron sputtering | [63] |
| $Zr_{60.14}Cu_{22.31}Fe_{4.85}Al_{9.7}Ag_3$ | Zr:Cu = 20<br>Zr:Fe = 22.5<br>Zr:Al = 10.6<br>Zr:Ag = 9.8 | Zr-Cu = −23; Zr-Fe = −25<br>Zr-Al = −44; Zr-Ag = −69<br>Cu-Fe = 13; Cu-Al = −1<br>Cu-Ag = 2; Fe-Al = −11<br>Fe-Ag = 28; Al-Ag = −4 | DC Magnetron sputtering | [64] |
| $Zr_{59}Cu_{20}Al_{10}Ni_8Ti_3$ | Zr:Cu = 20<br>Zr:Al = 10.6<br>Zr:Ni = 22.3<br>Zr:Ti = 9.6 | Zr-Cu = −23; Zr-Al = −44<br>Zr-Ni = −49; Zr-Ti = 0<br>Cu-Al = −1; Cu-Ni = 4<br>Cu-Ti = −9; Al-Ni = −22;<br>Ni-Ti = −35 | Arc-melting | [65] |
| $Fe_{40}Ni_{40}P_{14}B_6$ | Fe:Ni = 0.3<br>Fe:P = 14.5<br>Fe:B = 34.1<br>Ni:P = 14.8<br>Ni:B = 34.1 | Fe-Ni = −2; Fe-P = −39.5<br>Fe-B = −26; Ni-P = −34.5<br>Ni-B = −24; P-B = 0.5 | Induction melting, fluxing, re-melt, and quenching | [66] |
| $Fe_{50.26}B_{2.62}Si_{2.41}Cr_{23.86}Mo_{20.85}$ | Fe:B = 34.1<br>Fe:Si = 7.1<br>Fe:Cr = 0.6<br>Fe:Mo = 9.8 | Fe-B = −26; Fe-Si = −35<br>Fe-Cr = −1; Fe-Mo = −2<br>B-Si = −14; B-Cr = −31<br>B-Mo = −7; Si-Cr = −37<br>Si-Mo = −35; Cr-Mo = 0 | Atmospherically plasma-sprayed | [67] |
| Ti–Ni–Cu–Sn,<br>Ti–Ni–Cu–Sn–Be and<br>Ti–Ni–Cu–Sn–Be–Zr | Ti:Ni = 14.8<br>Ti:Cu = 12.6<br>Ti:Sn = 8.1<br>Ti:Be = 22.8<br>Ti:Zr = 9.6 | Ti-Ni = −35; Ti-Cu = −9<br>Ti-Sn = −21; Ti-Be = −30;<br>Ti-Zr = 0; Ni-Cu = 4<br>Ni-Sn = −4; Ni-Be = −4<br>Ni-Zr = −49; Cu-Sn = 7<br>Cu-Be = 0; Cu-Zr = −23<br>Sn-Be = 15; Sn-Zr = −43<br>Be-Zr = −43 | Injection casting | [68] |
| $Cu_{47}Ti_{34}Zr_{11}Ni_8$,<br>$Cu_{47}Ti_{33}Zr_{11}Ni_8Fe_1$ and<br>$Cu_{47}Ti_{33}Zr_{11}Ni_8Si_1$ | Cu:Ti = 14.3<br>Cu:Zr = 25.4<br>Cu:Ni = 2.6<br>Cu:Fe = 2.9<br>Cu:Si = 9.8 | Cu-Ti = −9; Cu-Zr = −23<br>Cu-Ni = 4; Cu-Fe = 13<br>Cu-Si = −19; Ti-Zr = 0<br>Ti-Ni = −35; Ti-Fe = −17<br>Ti-Si = −66; Zr-Ni = −49<br>Zr-Fe = −25; Zr-Si = −84<br>Ni-Fe = −2; Ni-Si = −40 | Copper mold casting | [69] |

## 3. Fabrication Techniques

Different fabrication routes can be adopted to provide randomness in the atomic orientation of the multicomponent systems. Metallic glasses are formed primarily by conventional casting or thermal-quenching techniques. Some novel fabrication techniques with their distinctive features will also be discussed in the following sections.

### 3.1. Liquid-Quenching Method

A significant fraction of bulk metallic-glass systems are formed by the liquid-to-solid transition technique [70–72]. To form a multicomponent metallic glass, metallic species are converted to a liquid by heating above the melting/liquidus temperature and later solidified by fast cooling. Thus, the liquid-quenching step requires a wide range of cooling rates. Based on the required cooling rate, the solidification technique can be classified into several groups, which are listed in Table 2. The extreme cooling rate in quenching techniques is found to be reached by distributing a thin layer of liquid in contact with a thermally conductive substrate (metal or sapphire) [25].

**Table 2.** Metallic-glass systems are fabricated by different solidification techniques and associated cooling rates.

| Solidification Technique | Cooling Rate (K/s) | Fabricated System | Ref. |
|---|---|---|---|
| Conventional die casting | $10^1$–$10^3$ | $Zr_{46.75}Ti_{8.25}Cu_{7.5}Ni_{10}Be_{27.5}$, Zr–Al–Cu, $La_{55}Al_{25}Ni_{20}$, $Mg_{80}Cu_{10}Y_{10}$, | [6,12,53,73] |
| Melt spinning | $10^5$–$10^6$ | $Fe_{57.2}Co_{30.8}Zr_{7-x}Hf_xB_4Cu_1$ (x = 3, 5, and 7), $Zr_{61}Cu_{17.5}Ni_{10}Al_{7.5}Si_4$ | [74–76] |
| Liquid splat-quenching | ∼$10^9$–$10^{10}$ | $Zr_{46.7}Ti_{8.3}Cu_{7.5}Ni_{10}Be_{27.5}$ Au-Si, $Au_{0.778}Ge_{0.138}Si_{0.084}$ | [2,77–79] |
| Pulsed laser quenching | ∼$10^{12}$–$10^{13}$ | Cu-Ti-Zr, Cu-Ti, $Ni_xNb_{100-x}$ | [80–83] |
| Nano calorimetry | $10^4$–$10^6$ | Au–Cu–Si | [84,85] |

The quenching rate is defined with reference to the heat-transfer capacity between the liquid and the substrate as well as on the thickness and thermal conductivity of the liquid layer [25]. For example, to quench the liquid in a conventional die-casting method requires a cooling rate within the range of 10 K/s to $10^3$ K/s [53]. Two temperature transitions occur in the formation of glass during quenching. The first transition of temperature is the liquidous temperature and the other one is the glass-transition temperature. According to Greer [74], decreasing the difference between these two temperatures favors the formation of metallic glass. In addition, glass formation is more likely to occur with rapid cooling of the liquid [74]. To extensively study the effect of the cooling rate, Kuball et al. [86] investigated the crystallization behavior of the $Al_{86}Ni_8Y_6$ system. As illustrated in Figure 1, Kuball et al. [86] concluded that apart from the material composition, considerably high cooling rates can limit the formation and growth of the intermetallic compound ($Al_{23}Ni_6Y_4$) and solid solution phase (α-Al) and form a monolithic metallic glass [86].

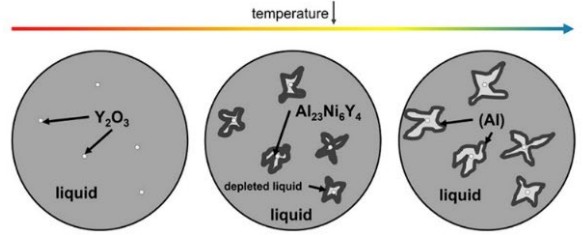
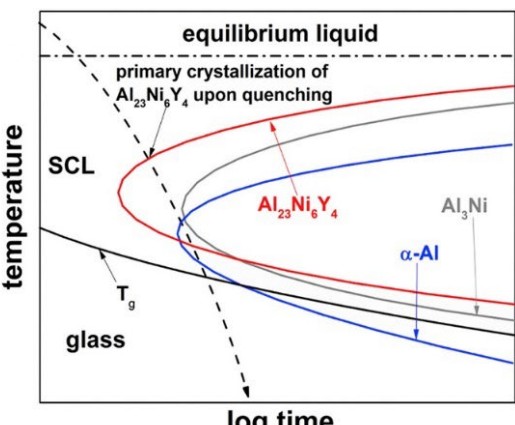

**Figure 1.** The effect of temperature during the solidification of $Al_{86}Ni_8Y_6$ is shown [86]. Adding $Y_2O_3$ particles into the liquid melt forms an $Al_{23}Ni_6Y_4$ intermetallic. Finally, the fcc α-Al solid solidifies at the interface of the $Al_{23}Ni_6Y_4$ intermetallic. A time–temperature–transformation (TTT) diagram shows the influence of heating and cooling rates on the crystallization behavior [86].

Although metallic-glass systems produced through rapid quenching techniques are typically multicomponent, a monoatomic metallic-glass system can be fabricated using extremely high cooling rates ($10^{14}$ K/s) [87]. Zhong et al. brought two Ta nano-tips into contact and melted them using short square-wave electric pulses (~3.7 ns and 0.5–3 V) [87]. As shown in Figure 2, they showed that the liquid was vitrified with a very high cooling rate to form a Ta metallic glass with a length and thickness on the nanometer scale [87]. However, the requirement of enormously high cooling rates to circumvent crystallization sets a severe restriction on the dimensions and geometry of the fabricated samples reachable using this process.

### 3.2. Welding of Metallic Glasses

The commercial production of metallic-glass systems is hampered by the requirement of a high cooling rate to fabricate a large-sized sample via casting. To eliminate the size constraint and fabricate bulk amorphous systems, a welding approach has been explored to join pieces of metallic glasses without compromising their excellent physical properties. The fabrication of a metallic glass composite (as shown in Figure 3) via this technique extends the application range of metallic-glass systems [88].

Different liquid-state joining techniques, such as electron-beam welding [89], laser-beam [88,90] welding, the pulse-current method [91], and gas tungsten arc welding [92] have been studied for different multicomponent metallic glasses. Liquid-state joining processes exhibit high energy density and deep weld penetration, which are helpful compared to conventional welding techniques [93]. However, phase stabilization in the heat-affected and weld-fusion zones to form a completely disordered solid remains a research challenge [94]. Among the liquid-state welding techniques, electron- and laser-beam welding have been extensively investigated. For example, a study conducted by Wang et al. [94] on the formation of Ti-based ($Ti_{40}Zr_{25}Ni_3Cu_{12}Be_{20}$) metallic glass using laser welding showed that the microstructure of both the weld fusion zone (WFZ) and heat-affected zone (HAZ) are affected by the welding speed. In the WFZ, a high-speed welding of 10 m/min provides a high cooling rate of 780 K/s, which results in an amorphous microstructure. In the WFZ, the lower welding speed would cause a high temperature exposure for a longer period of time, leading to crystallization in the microstructure. However, at higher welding speed, the HAZ would experience a fast temperature drop below the crystallization temperature, suppressing the crystallization [94]. The reason for this is attributed to the high heating rate; decreasing the welding speed tends to suppress crystallization [94].

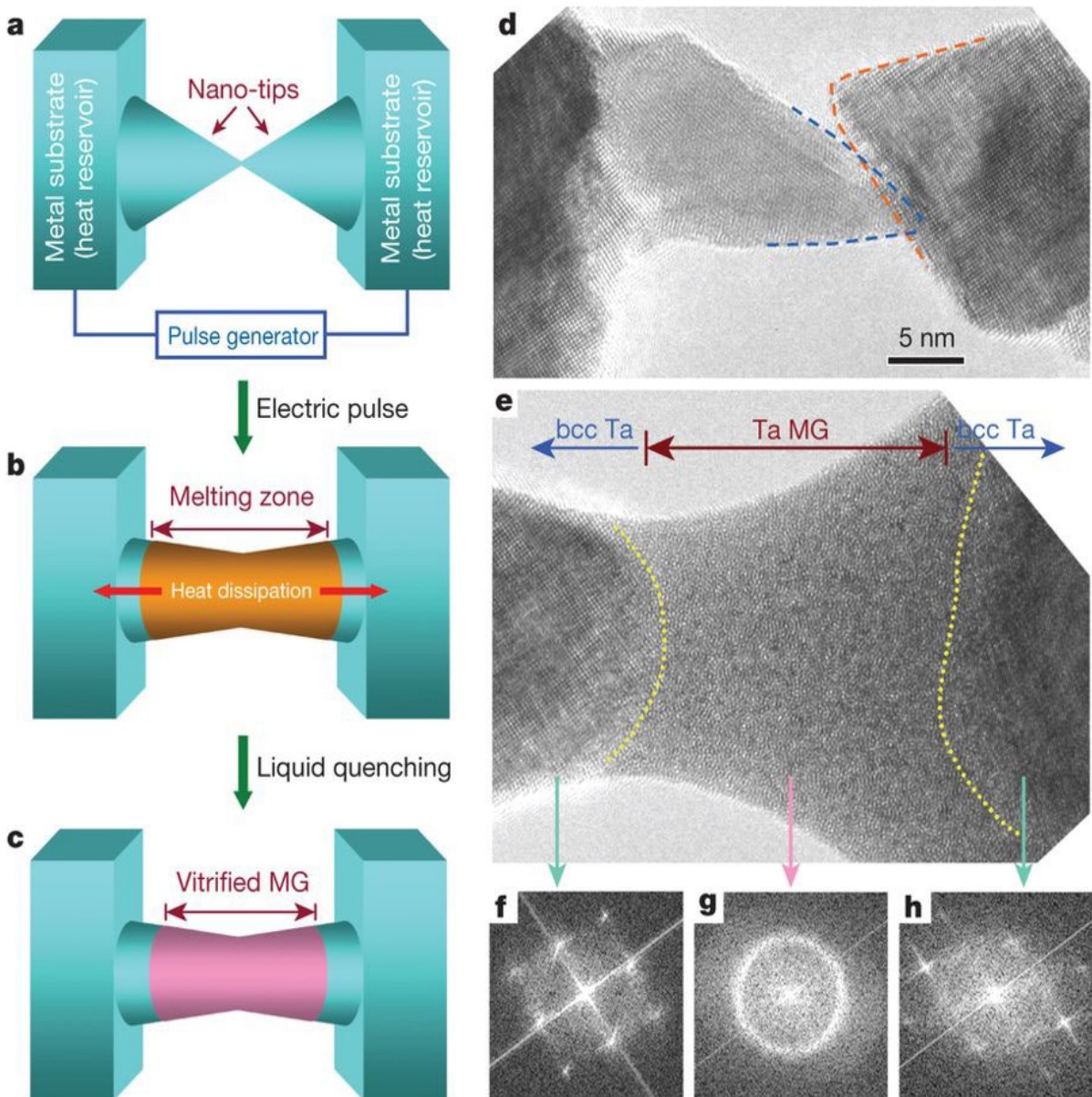

**Figure 2.** (**a**–**c**) Schematic drawing of the experimental configuration of an ultrafast liquid-quenching process [87]. (**a**) The process begins by bringing two protruded nano-tips in contact, which are melted by the application of an electric pulse; (**b**) heat is dissipated through the bulk substrates, and (**c**) the melting zone transforms into monatomic MGs due to liquid quenching. (**d**) High-resolution transmission electron microscope (HRTEM) image showing two contacting Ta nano-tips before the application of the electric pulse. The boundary of the two-contacting nano-tips are shown using the dotted lines (**e**) HRTEM image showing vitrified Ta MG after the application of the electric pulse. (**f**–**h**) Fast Fourier transformations confirming a fully vitrified region in the middle (**g**) bounded by two crystalline substrates.

Solid-state joining techniques, including friction [95], explosion [96], thermoplastic deforming [97], and ultrasonic welding [98] methods, are also being investigated. These processes produce relatively low temperatures and are found to provide excellent joining without crystallization [99]. Among the supercooled-liquid welding techniques, friction is widely used for joining metallic-glass systems and has been studied by several groups [95,100]. The idea of joining metallic-glass systems using friction stir welding originated from the metal-extrusion technique. In metal extrusion, the strong bond within

powder particles is established during the consolidation of the powder, due to superplastic deformation of the supercooled liquid [101]. Kamamura et al. [102] successfully joined $Pd_{40}Ni_{40}P_{20}$ BMG using friction, taking advantage of the superplasticity of its supercooled liquid. The welded system was found to have properties similar to the parent system that was fabricated through the quenching technique. In addition, Wang et al. [103] studied the effect of friction time and rotational speed to maintain the glassy state within the interface and concluded that there is a critical time for any rotational speed to keep the interface below the crystallization temperature.

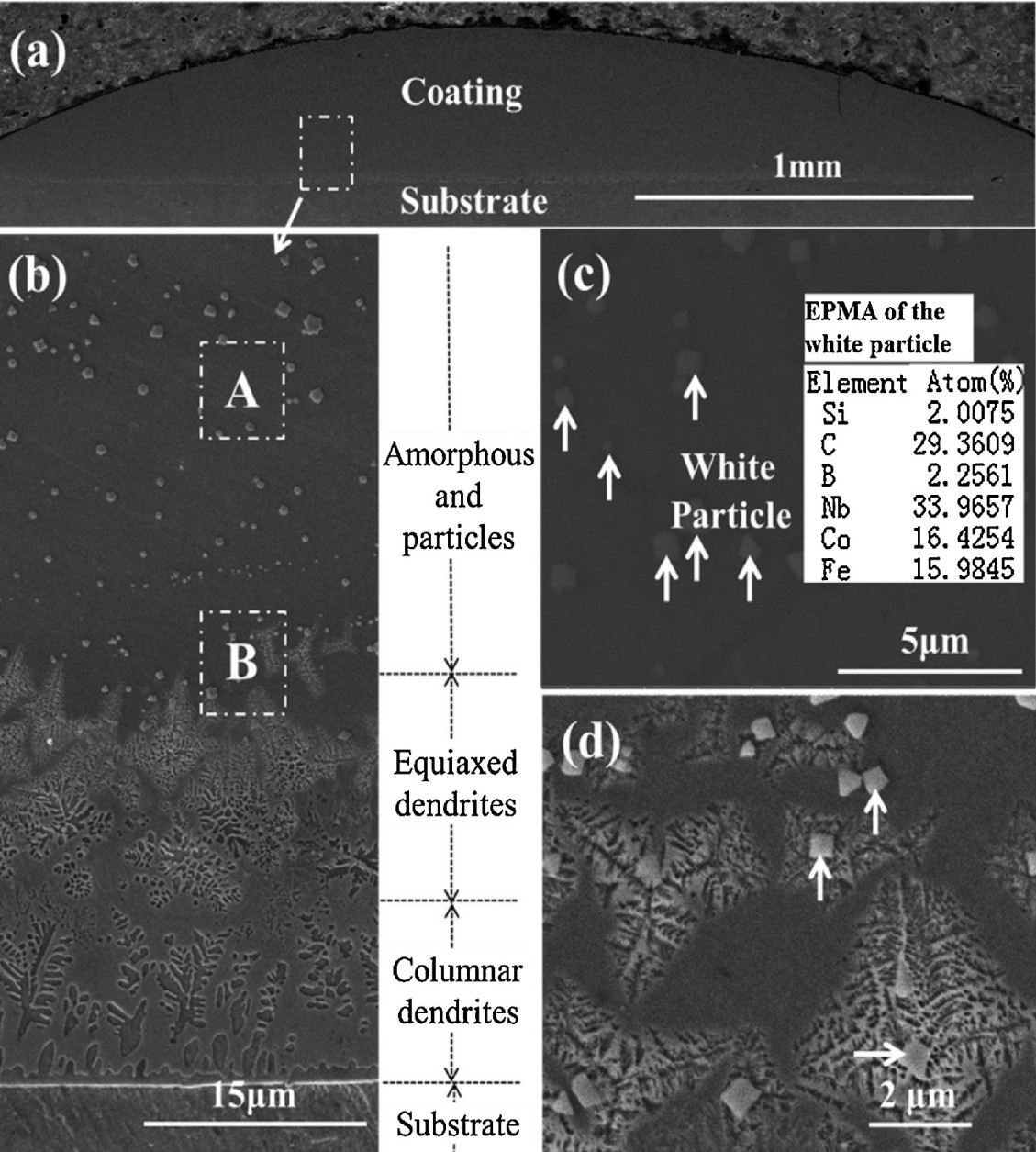

**Figure 3.** Scanning electron microscope (SEM) images of (**a**) the microstructure of the cross section of the $Fe_{34}Co_{34}B_{20}Si_5C_3Nb_4$ amorphous composite coating, (**b**) that near the substrate/coating interface, (**c**) a magnified view of rectangular region A, and (**d**) a magnified view of rectangular region B. The inset in (**c**) lists the electron probe microanalysis (EPMA) data of the white particle [88].

Using welding techniques to join disordered solids may allow the production of large metallic-glass systems, which can solve the issue with the requirement of a very high

cooling rate. However, more research on this field will determine the intrinsic mechanism to join different metal-based MG systems. Different metallic-glass systems joined by liquid- and solid-state techniques are listed in Table 3.

**Table 3.** Metallic-glass systems joined via different welding techniques.

| Metallic Glass System | Welded to | Welding Technique | Parameters | | | Ref. |
|---|---|---|---|---|---|---|
| | | | Thickness (mm) | Power (kW) | Scanning Speed (mm/s) | |
| **Liquid-state welding** | | | | | | |
| $Zr_{41}Be_{23}Ti_{14}Cu_{12}Ni_{10}$ | $Zr_{55}Al_{10}Ni_5Cu_{30}$ | Electron-beam | 3.5 | 9 | 33 | [104] |
| $Zr_{41}Ti_{14}Cu_{12}Ni_{10}Be_{23}$ | Polycrystalline Zr metal | Electron-beam | 3 | 9 | 33 | [105] |
| $Zr_{41}Be_{23}Ti_{14}Cu_{12}Ni_{10}$ | Ti metal | Electron-beam | 3 | 9 | 66 | [106] |
| $Zr_{41}Be_{23}Ti_{14}Cu_{12}Ni_{10}$ | Stainless steel | Electron-beam | 2 | 9 | 66 | [107] |
| $Zr_{45}Cu_{48}Al_7$ | $Zr_{45}Cu_{48}Al_7$ | Laser | 1 | 1.2 | 33–133 | [90] |
| $Cu_{54}Ni_6Zr_{22}Ti_{18}$ | $Cu_{54}Ni_6Zr_{22}Ti_{18}$ | Pulsed Nd:YAG | 6 | 1.5 | 0.33 and 1 | [108] |
| $Zr_{55}Al_{10}Ni_5Cu_{30}$ | 304 austenitic stainless steel | Fiber-laser | 9 | 2–10 | 1.2 | [109] |
| $(Zr_{53}Cu_{30}Ni_9Al_8)Si_{0.5}$ | $(Zr_{53}Cu_{30}Ni_9Al_8)Si_{0.5}$ | Pulsed Nd:YAG | 1 | 1.3–1.7 | 1 | [110] |
| $Pd_{43}Cu_{27}Ni_{10}P_{20}$ | $Pd_{43}Cu_{27}Ni_{10}P_{20}$ | Pulsed laser beam | 1 | 0.750–1.125 | 0.33 | [111] |

| Metallic glass system | Welded to | Welding process | Parameters | | |
|---|---|---|---|---|---|
| | | | Rotational speed (rpm) | Time (s) | |
| **Solid-state welding** | | | | | |
| $Ti_{40}Zr_{25}Ni_3Cu_{12}Be_{20}$ | $Ti_{40}Zr_{25}Ni_3Cu_{12}Be_{20}$ | Friction | 1800–2200 | 5–7 | [103] |
| $Zr_{41.5}Ti_{13.8}Cu_{12.5}Ni_{10}Be_{22.5}$ | $Zr_{41.5}Ti_{13.8}Cu_{12.5}Ni_{10}Be_{22.5}$ | Friction | 2700 | 0–35 | [112] |
| $Zr_{55}Al_{10}Ni_5Cu_{30}$ | $Zr_{55}Al_{10}Ni_5Cu_{30}$ | Friction | 1800 | 0.4–1.0 | [113] |
| $Pd_{40}Ni_{40}P_{20}$ | $Pd_{40}Cu_{30}Ni_{10}P_{20}$ | Friction | 6000 | 0.2 | [114] |

### 3.3. Additive Manufacturing

In recent year, additive manufacturing (AM) has drawn much attention from the manufacturing industries and academia due its unique features that can significantly affect the microstructure and properties of engineering materials. This emerging additive manufacturing technology is also known as three-dimensional (3D) printing, in which a layer-by-layer fabrication process is used to build a 3D object directly from a CAD file [115]. Through additive manufacturing, it is possible to produce geometrically intricate designs, which would not be possible using conventional manufacturing techniques [115]. It is also possible to improve and customize material properties according to desired applications via the proper selection of process parameters during fabrication. Metallic-glass systems have also been produced by different additive manufacturing techniques [116–119]. Selective laser melting (SLM) is a branch of additive manufacturing, which is used for the formation of disordered solids [120–122] and does not require precise molds for the fabrication of complex and large-scale parts [123]. For example, SLM has been used to fabricate metallic-glass systems, which has been found to be beneficial as a composite matrix for biomedical applications [124,125].

In SLM, metal powder is placed on a metal plate and melted using a laser. The melt freezes rapidly and bonds with the metal plate. Additional layers of powder are then added as the melting and freezing steps are continued [123]. The overall process occurs in a closed chamber, where argon or helium is used to create an inert atmosphere to minimize contamination. The metal plate in the SLM process serves as the support for the final part and is usually iron (Fe), titanium (Ti), aluminum (Al) or nickel (Ni) [122,126] due to their outstanding mechanical stabilities and heat-dissipation characteristics [122]. Both the heating and cooling rates during the SLM process require the careful selection of parameters (laser power density, spot size, hatch spacing and scan speed) for the formation of a metallic glass [127]. The energy input into the molten powder is controlled by changing the spot size, the power of the laser, and the dwell time [128]. Optimizing these parameters can minimize the melt puddle, which is found to facilitate the glass formation [128].

The effects of scan speed and laser power to form a disordered structure have been studied by Jung et al. [129] for Fe-based ($Fe_{68.3}C_{6.9}Si_{2.5}B_{6.7}P_{8.7}Cr_{2.3}Mo_{2.5}Al_{2.1}$) metallic glass. Cross-section images of the resulting samples are shown in Figure 4. Jung's study indicates that optimal energy transfer and high relative energy densities are helpful for forming metallic glass, which is achieved by using low scan speed and high laser power [129]. However, a study conducted by Li et al. indicates that using too high a laser energy density can introduce some crystallization into a system [130]. The impressive part of the layer-by-layer fabrication process of AM is that a refined microstructure and outstanding mechanical features can be achieved using SLM [129] as compared to samples produced via conventional fabrication techniques. However, crystallization is difficult to avoid due to the thermal processes involved in the SLM technique [127]. Moreover, thermal stress can induce cracks within the samples, which can effectively be mitigated by decreasing the energy density during the fabrication process [131]. Different metallic-glass systems fabricated using additive manufacturing process are listed in Table 4, along with some of the associated fabrication process parameters.

**Table 4.** Metallic-glass systems produced via additive manufacturing process.

| Materials Systems | Laser Power (W) | Scanning Speed (mm/s) | Hatch Spacing (μm) | Ref. |
|---|---|---|---|---|
| $Zr_{55}Cu_{30}Ni_5Al_{10}$ | 240 | 1200 | 100 | [120] |
| $Al_{86}Ni_6Y_{4.5}Co_2La_{1.5}$ | 120 | 750 | 100 | [132] |
| $Al_{85}N_5Y_6Co_2Fe_2$ | 200 | 625 | 150 | [131] |
| $Ti_{47}Cu_{38}Zr_{7.5}Fe_{2.5}Sn_2Si_1Ag_2$ | 60 | 2000 | 140 | [133] |
| $Zr_{52.5}Cu_{17.9}Ni_{14.6}Al_{10}Ti_5$ | 30–120 | 250–2000 | 100–200 | [134] |
| $Fe_{74}Mo_4P_{10}C_{7.5}B_{2.5}Si_2$ | 320 | 3470 | 124 | [123] |
| $Zr_{52.5}Ti_5Cu_{17.9}Ni_{14.6}Al_{10}$ | 200 | 500 | 150 | [130] |
| $Fe_{43.7}Co_{7.3}Cr_{14.7}Mo_{12.6}C_{15.5}B_{4.3}Y_{1.9}$ | 150–350 | 200–1000 | - | [135] |
| $Fe_{54.35}Cr_{18.47}Mn_{2.05}Mo_{13.93}W_{5.77}B_{3.22}C_{0.90}Si_{1.32}$ | 220–380 | 2000 | 90 | [136] |
| $Zr_{50}Ti_5Cu_{27}Ni_{10}Al_8$ | 200 | 13.3 | - | [137] |

### 3.4. Powder Densification Technique

Metallic glasses produced by quenching techniques have size limitations (mm to cm) due to the requirement of very high cooling rates. A potential solution to avoid the issue of the cooling rate is the use of a fabrication route that consolidates amorphous powder. In this method, mechanical alloying [138,139] or high-pressure argon-gas atomization [140,141] is usually adopted to produce amorphous powders. Mechanical alloying is a simple, room-temperature method; however, it is a lengthy process. On the other hand, the high-pressure gas atomization technique requires an extreme experimental condition [142].

The amorphous powders are densified using other processes, such as extrusion methods [143–145], cold or hot pressing [146,147], spark-plasma sintering [139,148], and injection [149]. These densification processes are classified based on their operating temperature and time. For example, cold pressing [146] or equal channel angular extrusion [144] is performed at room temperature, whereas warm extrusion [150], hot pressing [147], injection [149], and spark-plasma sintering [142] require high temperatures. Among all these techniques, spark plasma sintering is popular due to densification capability within a short span of time [127]. Moreover, the spark plasma sintering process combined with dynamic pressing and fast heating is advantageous for the formation of a disordered solid, even for the marginal glass formers [127,151]. However, there is a possibility of crystallization if thermal stability is not controlled during the densification process [142].

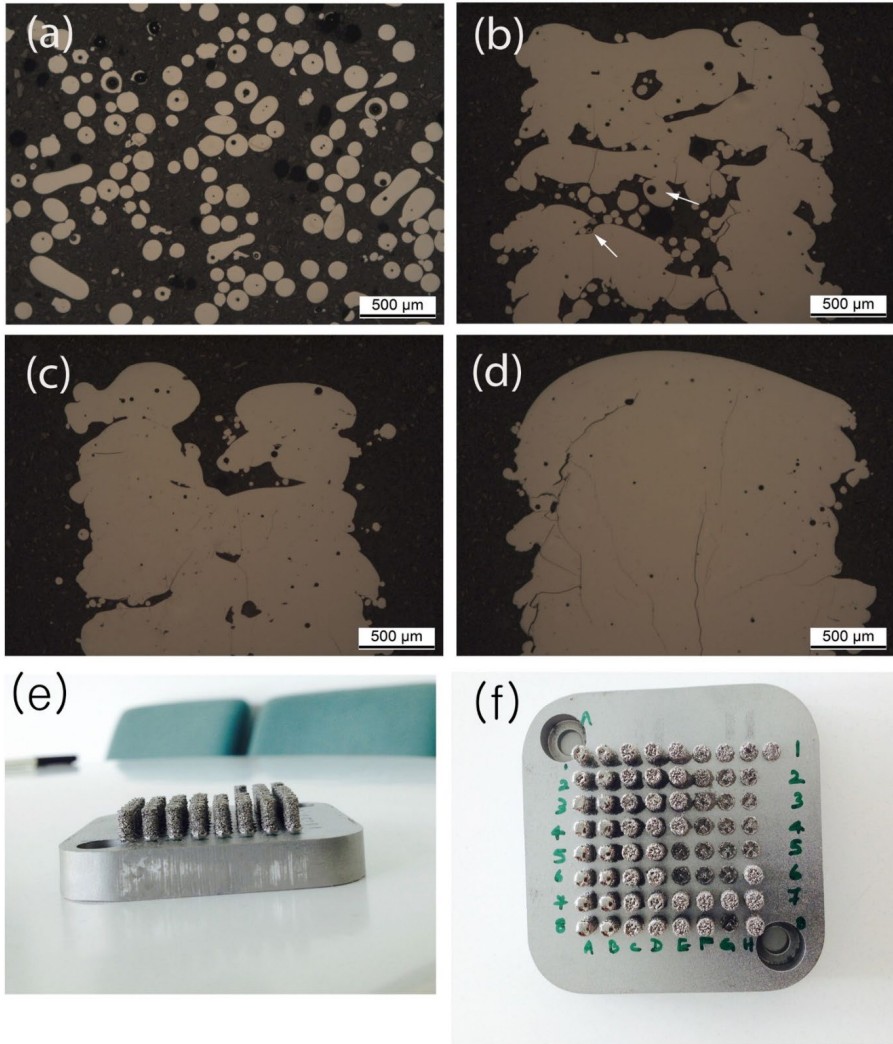

**Figure 4.** Cross-sectional optical microscope images showing (**a**) atomized powders on a substrate. Samples prepared using selective laser melting (SLM) with (**b**) v = 2500 mm/s and P = 300 W, (**c**) v = 2500 mm/s and P = 340 W, and (**d**) v = 1500 mm/s and P = 340 W. (**e**) Side- and (**f**) top-view of the final SLM samples. The observed micro-pores upon using v = 2500 mm/s and P = 300 W are shown with white-arrows in (**b**). [129].

Cardinal et al. [142,152] extensively studied a Cu-based metallic glass ($Cu_{50}Zr_{45}Al_5$) fabricated via the spark plasma sintering method. In this study, planetary ball milling is used to make crystalline Zr (as shown in Figure 5a) and amorphous CuZrAl powders (Figure 5b). The obtained pure crystalline Zr (20% vol.) powder is homogeneously mixed with the amorphous CuZrAl powder (Figure 5c). The mixed powders (crystalline Zr and amorphous CuZrAl) are densified using the spark plasma sintering technique. As illustrated in Figure 6, the sintering temperature was 420 °C with an imposed pressure of 600 MPa. The application of high pressure and a fast-heating rate help to create a stronger bond between the powder grains and result in an amorphous solid [142].

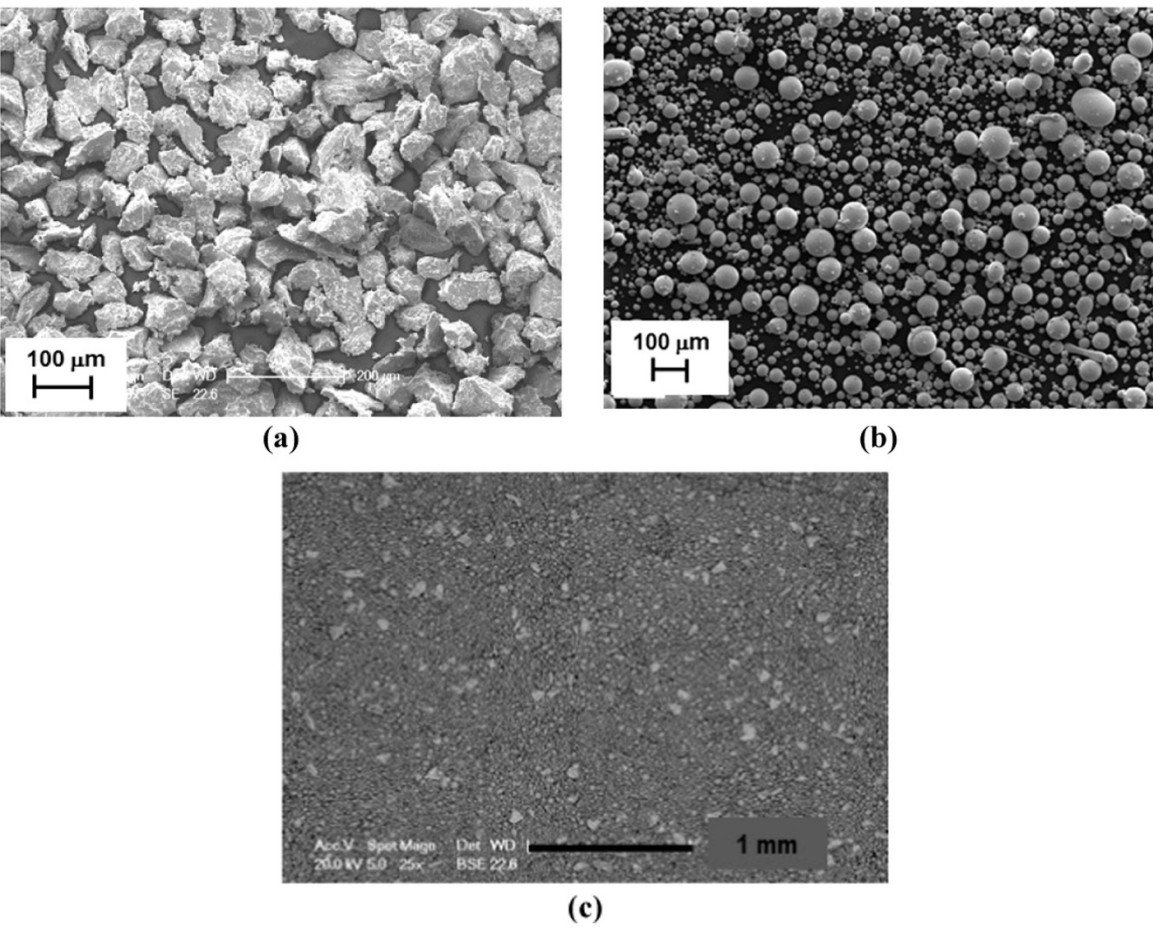

**Figure 5.** Illustration of (**a**) the crystalline Zr–powder, (**b**) amorphous CuZrAl powder, and (**c**) a mixture of 20% vol of Zr-powder with amorphous powder produced using the ball-milling technique [152].

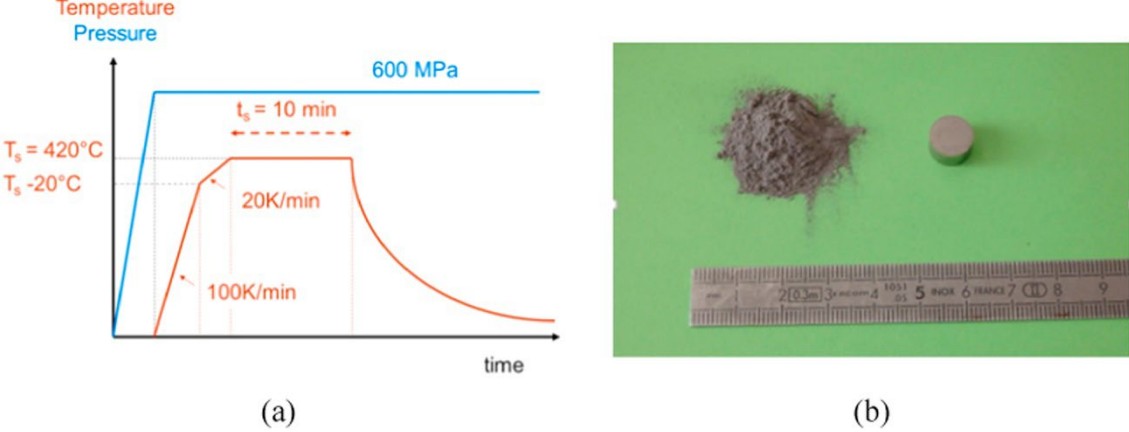

**Figure 6.** (**a**) Temperature and pressure cycle of a spark plasma sintering (SPS) densification technique, and (**b**) cylindrical sample produced from the powder after densification [152].

The capability of porous metallic-glass systems to achieve outstanding mechanical properties (e.g., high strength, large elastic stains) and corrosion resistance has raised much research interest [153]. Conventional casting is found to hinder the successful formation of porous metallic glass structures due to the size constraints and the high contamination tendency [154]. In addition, spark plasma sintering alongside densification techniques are found to be a successful alternative to the methods that require very high cooling

rates [155–157]. Different metallic-glass systems formed through the powder densification techniques are listed in Table 5.

**Table 5.** Metallic-glass systems produced via powder metallurgy.

| Name of the System | Powder Production Technique | Densification Process | Ref. |
| --- | --- | --- | --- |
| $Ni_{59}Zr_{20}Ti_{16}Si_2Sn_3$ | Argon gas atomization | Warm extrusion | [143] |
| $Cu_{50}Ti_{32}Zr_{12}Ni_5Si_1$ | High-pressure gas-atomization | Equal channel angular extrusion | [144] |
| $Mg_{65}Cu_{25}Gd_{10}$ | Mechanically milled | Spark plasma sintering | [138] |
| $Ni_{53}Nb_{20}Ti_{10}Zr_8Co_6Ta_3$ | Mechanical alloying | Spark plasma sintering | [139] |
| $Al_{82}La_{10}Ni_4Fe_4,$ | Mechanical alloying | Spark plasma sintering | [158] |
| $Ni_{52.5}Nb_{10}Zr_{15}Ti_{15}Pt_{7.5}$ | Argon gas atomization | Spark plasma sintering | [159] |
| $Zr_{55}Cu_{30}Al_{10}Ni_5$ | Argon gas atomization | Spark plasma sintering | [155] |
| $Ni_{59}Zr_{15}Ti_{13}Si_3Sn_2Nb_7Al_1$ | Gas-atomization | Spark plasma sintering | [160] |
| $(Fe_{0.72}B_{0.24}Nb_{0.04})_{95.5}Y_{4.5}$ | Gas-atomization | Spark plasma sintering | [161] |
| $Fe_{73}Si_7B_{17}Nb_3$ | Argon gas atomization | Spark plasma sintering | [162] |
| $Fe_{48}Cr_{15}Mo_{14}Y_2C_{15}B_6$ | Argon gas atomization | Spark plasma sintering | [163] |
| $Fe_{67}Co_{9.5}Nd_3Dy_{0.5}B_{20}$ | Mechanically rotor-milled | Spark plasma sintering | [164] |
| $Ti_{50}Cu_{23}Ni_{20}Sn_7$ | Mechanically milled | Spark plasma sintering | [165] |

### *3.5. Magnetron Co-Sputtering*

Vapor quenching techniques, such as evaporation and magnetron co-sputtering, are a non-equilibrium vapor-to-solid transition method, which is often used to produce multicomponent metallic-glass systems [13]. In magnetron co-sputtering, the growth of the atoms on a substrate is governed by two competing factors, e.g., thermodynamics and kinetics [13]. When the energies of the arriving atoms were compared, the sputtered atoms were found to have average energy that is two orders of magnitude higher than that of the atoms in the evaporation technique [13]. Thus, from a thermodynamics perspective, the effective quench rate of the atoms is slower in sputtering than in evaporation [166]. However, from a kinetic point of view, the magnetron-sputtering process is specified in terms of momentum [13], and the targets with heavy atoms possess a higher momentum that facilitate the solidification of the atoms in a metallic glass system [167].

Sputtering systems are typically equipped with multiple guns, which are used to deposit materials on a substrate. Various fabrication parameters, such as the angle and power of each gun, gaseous environment, compositions of the targets, angle and rotation of the substrate, etc., can be carefully tuned to obtain optimized compositions of multicomponent metallic-glass systems [127]. By changing the aforementioned parameters, a compositional library of a metallic glass system can be synthesized for different engineering applications (as shown in Figure 7) [168]. Moreover, the slower cooling rate of sputtering and lack

of nucleation sites of crystallization lead to properties that often emerge simultaneously in a single metallic-glass system. Bouala et al. [169] introduced Ag in Zr-Cu to form a compositional library of a Zr-Cu-Ag metallic glass system and studied the glass-forming ability, mechanical properties, electrochemical properties, and antibacterial activity of the fabricated systems for potential bio-medical applications. Among all the compositions, $Zr_{73}Cu_{16}Ag_{11}$ metallic glass was found to exhibit optimum mechanical and electrochemical properties with enhanced antibacterial activity [169]. Metallic-glass systems produced by combinatorial development to achieve different properties are listed in Table 6.

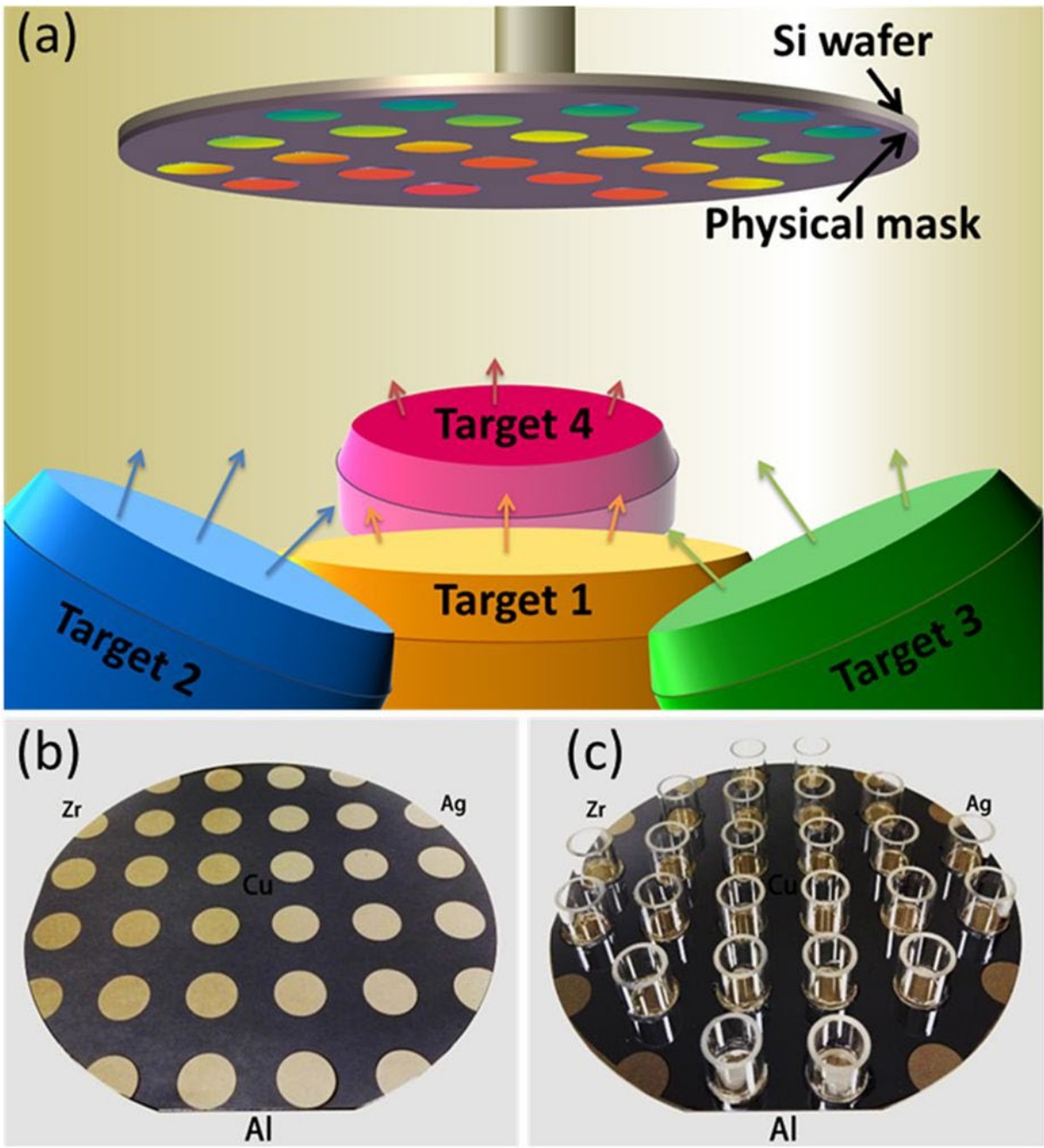

**Figure 7.** (**a**) Schematic illustration of magnetron co-sputtering of the Zr-Cu-Al-Ag system, (**b**) deposited films on the Si wafer where the compositional variation of the elements are obtained by tilting the guns at different angles, and (**c**) antibacterial activity analysis of the deposited films [168].

**Table 6.** Parameters associated with the magnetron-sputtering technique used to fabricate various metallic-glass systems.

| System | Sputtering Type | Parameters | | | | | | | Ref. |
|---|---|---|---|---|---|---|---|---|---|
| | | DC Power (W) | RF Power (W) | Base Pressure (Pa) | Working Pressure (Pa) | Target-Substrate Distance (mm) | Sputtering Rates (nm/min) | Thickness ($\mu$m) | |
| Zr–Ni–Al–Si | Reactive magnetron co-sputtering | Zr: 700 Ni: 120 | Al: 240 Si: 300 | $2 \times 10^{-4}$ | 0.4 | 250 | 2.1–6.1 | - | [61] |
| $Zr_{61}Al_{17.5}Ni_{10}Cu_{17.5}Si_4$ | DC plus magnetron sputtering | 30 | - | $7.9 \times 10^{-4}$ | 0.53 | 60 | - | 0.5 | [170] |
| ZrCuAl | RF magnetron sputtering | - | Zr: 225 Cu: 50 Al: 26 | $6.6 \times 10^{-5}$ | 1.3 | 165 | - | 2 | [171] |
| $Zr_{55}Cu_{31}Ti_{14}$ | DC co-sputtering | 300 | - | $6.6 \times 10^{-5}$ | - | - | - | 2–3 | [172] |
| Zr-Pd | RF magnetron sputtering | - | 70 | $2-4 \times 10^{-4}$ | 2 | 70 | 6.6 | - | [173] |
| ZrCuAlAg | DC magnetron sputtering | 15–30 | - | - | 0.4 | 60 | 7.6–14.5 | 0.2 | [62] |
| $Zr_{47}Cu_{31}Al_{13}Ni_9$ | RF magnetron sputtering | - | 100 | - | 0.27 | - | - | 0.2 | [174] |
| $Al_{48}Ag_{37}Ti_{15}$ | Magnetron sputtering | - | - | $6.6 \times 10^{-5}$ | 0.4 | - | - | 0.5 | [59] |
| Ta-Ti-Zr-Si | DC co-sputtering | - | - | $6.6 \times 10^{-5}$ | - | - | - | 0.6 | [175] |
| $Zr_{60.14}Cu_{22.31}Fe_{4.85}Al_{9.7}Ag_3$ | DC magnetron sputtering | 120 | - | $2 \times 10^{-4}$ | 0.65 | - | - | 0.53 | [64] |

## 4. Distinctive Properties

Like conventional crystalline materials, the properties of metallic-glass systems also originate from their microstructure, and the potential of metallic-glass systems as functional materials is an area of active research. The following sections summarize the structural, surface, mechanical, thermal, electrochemical, and magnetic characteristics of various metallic-glass systems that would be essential to solidify our understanding of these unique materials.

### 4.1. Structural and Surface Properties

Grain boundaries and secondary phase precipitates are the two common structural features that are absent in metallic-glass systems [176,177]. The homogeneous microstructure of metallic glass matrices contributes to their enhanced electrochemical and mechanical properties [35,177,178]. Furthermore, a better surface condition and minimal surface roughness are found to be useful in various engineering applications [58,179,180].

Various mathematical models have been proposed to explain the atomic structure of metallic-glass systems [181,182]. Among these models, Bernal's dense random packing model is useful in describing the monoatomic amorphous structures of a short-range order [183]. However, the model is unable to explain the structures of binary metallic glasses [184]. In this regard, the model suggested by Gaskall [185] is rather useful but lacks proper experimental evidence. The model proposed by Miracle [186] considered a medium-range order (MRO) of metallic-glass systems with face-centered cubic (fcc) atomic packing. Other studies also considered a short- to medium-range order (MRO) for modeling different metallic-glass systems [187–191]. For example, the model suggested by Lee et al. [181] considered both short- and medium-range orders of binary Cu-Zr metallic glass. The short-range order of the $Cu_{65}Zr_{35}$ system exhibited a high packing density and a low free volume, whereas for the medium-range order, the majority of the icosahedra were found to exist in linked clusters. The icosahedra connected in the medium-range order resulted in the lowest average potential and atomic volume, which were found to provide more structural stability [192]. The structural modeling of metallic-glass systems postulates efficient packing around the solute and solvent atoms, and lower densities of the amorphous systems compared to their crystalline counterpart [193,194]. The hypothesis was later verified experimentally by Battezzati and Baricco, who reported lower molar volumes of different metallic-glass compositions [195]. Mukherjee et al. reported that the composed icosahedral structures connected in an extensive network with higher atomic packing led to a higher viscosity and lower atomic volume of metallic-glass systems [196]. Both of

these two factors impact glass-forming ability and provide better structural properties of metallic-glass systems [197].

Different diffraction patterns and nano-scale characterization techniques have been carried out to analyze the disordered structure. The study conducted by Khan et al. [198] showed the microstructural analysis of a Zr-based (Zr-Ti-Fe-Al) metallic glass system using the HR-TEM technique. Figure 8 depicts the amorphous structure with embedded nano-crystallites, which was affirmed through selected-area electron diffraction (SAED) combined with fast Fourier transformation (FFT). The influence of inclusion on the properties of disordered solids was extensively examined by Liu et al. [199] by introducing different elements into the Cu-Ag system. It was found that the inclusion of Cr and Co in Cu-Ag improves electrical conductivity, with a slight compromise in strength, whereas the inclusion of Zr leads to a reverse trend. The relationship between the metallic glass composition, microstructure, and properties of various Zr-(Cu,Ag)-Al compositions was also reported by Jiang et al. and Chen et al. [62,197]. The synthesized metallic-glass systems are found to exhibit different physical and chemical properties with the change in sputtering power.

The composition of metallic glasses has been found to trigger different surface properties [197,199,200]. The increase in roughness with sputtering power indicates that under certain conditions, the power level can provide a composition with a better surface finish, whereas a higher deposition rate can be attributed to a higher surface roughness. Another study on Zr-based metallic glass (Zr–Ni–Al–Si) examined a broad range of compositions and nitrogen (0–17.7%) contents for improved physical and chemical features [61]. The inclusion of nitrogen of more than 7.1% was found to induce a higher level of crystallinity into the system, whereas the system without the inclusion of nitrogen was found to have a minimal amount of roughness. Moreover, thermally annealed metallic glass with an atomic composition of 60% Cu and 40% Zr exhibited an outstanding smooth surface, with fine grains and a dense microstructure [201]. In this study, thermal annealing was stated to impede the nucleation of nano-crystals due to the reduction in free volume in the atomic structure [201]. A study conducted by Ishii et al. also assessed the effect of thermal annealing in the atomic structure of metallic glass, where the structural analysis of the metallic glass system indicates the absence of pores with large open spaces, and it was predicted that the atomic rearrangement near the free volume does not occur due to the thermal treatment [202]. The study indicates that the metallic glasses may not consist of an absolute random structure; rather, the disordered state is influenced by the chemical order of the systems used for alloying [202]. Moreover, Zr-($Zr_{53}Cu_{33}Al_9Ta_5$)- and Cu-($Cu_{48}Zr_{42}Ti_4Al_6$)-based metallic glasses are found to exhibit a better surface condition, as the wettability of the surfaces indicates a significant reduction in the surface roughness from the crystalline materials [63]. A study conducted by Liu et al. [168] synthesized the compositional library for the Zr-based metallic-glass systems, and the disordered structural condition was found to exhibit lower roughness values compared to the crystalline materials.

### 4.2. Thermal Stability

The formation of metallic glass requires a proper understanding of thermodynamics, as the glass formation and observed material properties are interrelated. It is anticipated that the addition of specific elements in metallic-glass systems may often decrease the glass-forming ability [203]. A study conducted by Inoue et al. [204] indicated that the presence of Co, Cr, Fe, Mo, and V diminished the supercooled region due to a decrease in crystallization temperature ($T_x$), and, thus, reduced its glass-forming ability. The study determined that Hf inclusion in the Zr-based metallic glass kept the supercooled region unchanged, whereas the glass-phase area ratio extended with the inclusion of Ti, Nb, and Pd. Another study conducted by He et al. [203] analyzed the effect of Ta addition in the thermal stability of the supercooled region in a Zr-based system ($Zr_{52.25}Cu_{28.5}Ni_{4.75}Al_{9.5}Ta_5$) fabricated using an arc-melting process. The study indicated that the addition of 3.2% Ta improved the thermal stability of the metallic glass matrix, although it did not signif-

icantly improve its glass-forming ability. For the $Zr_{57}Nb_5Al_{10}Cu_{15.4}Ni_{12.6}$ metallic-glass system [205], the thermal stability of the system was tested by adding Mo, Nb, and Ta. It was observed that the addition of these species did not alter its thermal stability. Inoue and Zhang studied the thermal stability of Cu-Zr-Al metallic-glass systems [206] for a wide range of compositions, and it was found that the inclusion of Al (3–10%) increased the temperature difference ($\Delta T$) between the glass-transition temperature ($T_g$) and the crystallization temperature ($T_x$), which further stabilized the systems thermodynamically. For an Fe-based $Fe_{43}Cr_{16}Mo_{16}C_{15}B_{10}$ metallic glass [207], negative heat-of-mixing values were found within the range of 1–45 kJ mol$^{-1}$, which is within the range of stabilization for a supercooled liquid.

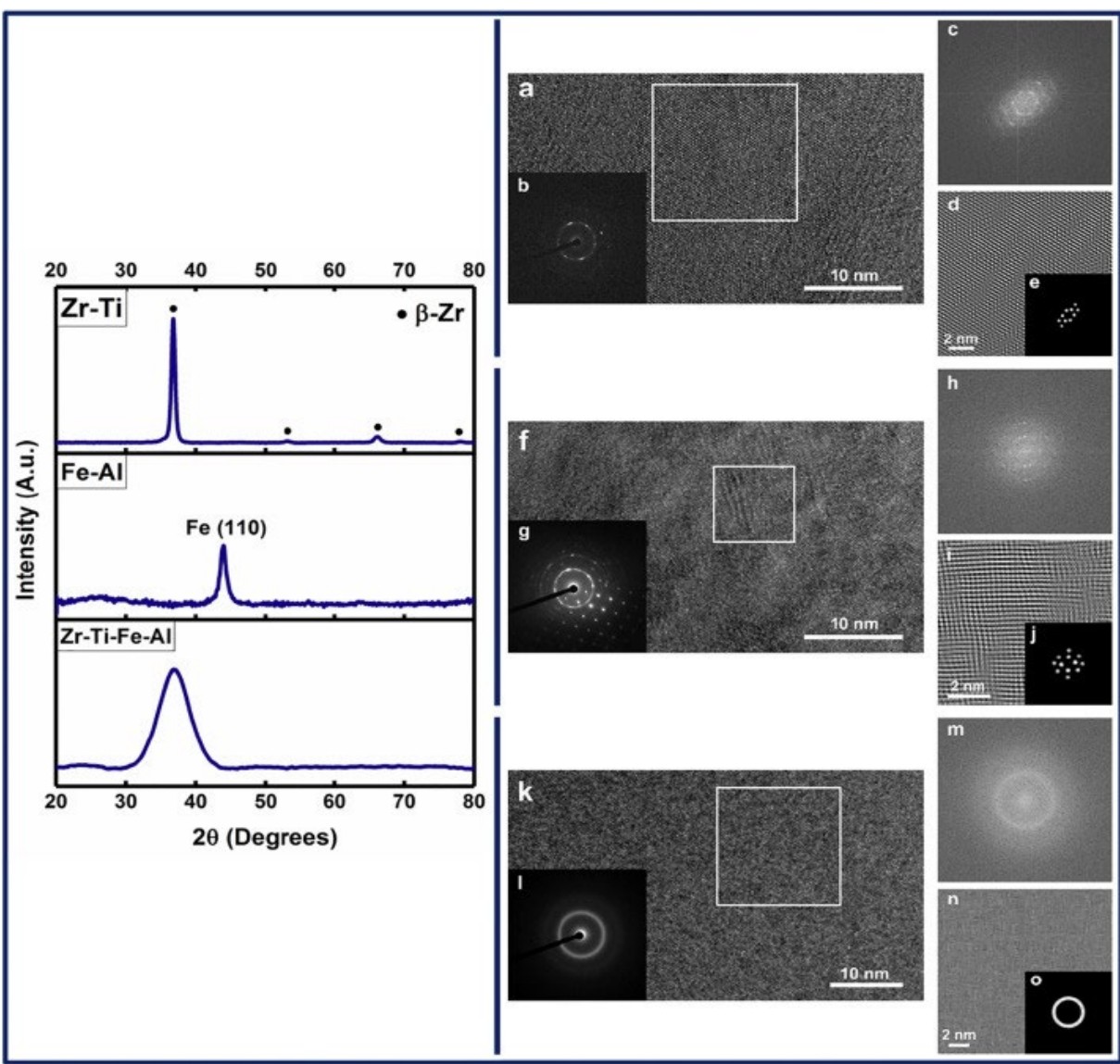

**Figure 8.** X-ray diffraction (XRD) and high-resolution transmission electron (HR-TEM) microscopy (**a**,**f**,**k**) images of Zr-Ti, Fe-Al and Zr-TI-Fe-Al systems, where the insets (**b**,**g**,**l**) show selected-area electron diffractions (SAED). The corresponding fast Fourier transformation (FFT), (**c**,**h**,**m**), inverse FFT (**d**,**i**,**n**), and filtered FFT (**e**,**j**,**o**) images of the white boxed region are also shown [198].

The empirical rules of Inoue [48] associated with the negative enthalpy between the atomic pairs are also related to the thermal stability of metallic-glass systems [207]. It has been observed that the systems that follow this particular rule usually form a denser and long-range configuration (by suppressing the nucleation for crystallization) with a relatively

lower atomic diffusivity and a higher viscosity [207,208]. For example, the formation of Fe-based metallic glass ($Fe_{43}Cr_{16}Mo_{16}C_{15}B_{10}$) with a higher $\Delta T$ is attributed to the low atomic diffusivity, nucleation, and growth reactions among the species [207]. Apart from the multicomponent systems, the glass-forming ability and thermal behavior of binary systems were also studied. For example, Zr-Cu systems of an extensive compositional range [50] were characterized using differential scanning calorimetry. The systems sputtered at a low-density discharge condition exhibited an optimum range of the stable supercooled region with the inclusion of 31–62 at% Cu, whereas the high-density discharge condition resulted in an inconsistent supercooled region. Finally, the thermal stability of the supercooled region, the atomic species, and the fabrication routes all govern the electrochemical [207], mechanical [203,205] and structural [202] responses of metallic-glass systems. Among different fabrication routes, the combinatorial method via sputtering has been reported to yield thermodynamically stable metallic-glass systems and has been adopted for easy glass formation [50,64,175,209].

### 4.3. Mechanical Properties

When metallic-glass systems are considered for structural, biomedical, defense, and aerospace applications, the evaluation of mechanical properties is essential [35,36,45,210,211]. A higher yield strength and increased elastic limit are the two properties that make metallic-glass systems distinct from commonly used crystalline materials [35,69]. Metallic-glass systems have been reported to have significantly higher yield strength and hardness values compared to 316 L stainless steel-, Ti-, and Zr-based alloys [35,75,212,213]. Moreover, a lower elastic modulus and comparatively higher elastic strain have also been reported for different metallic-glass systems, which are beneficial for applications where a reduction in stress concentration is vital [214–217]. The yield strength is the ability of a material to resist plastic deformation, and metallic-glass systems are found to have a higher yield strength [218]. Metallic-glass systems also exhibit higher wear resistance compared to crystalline materials. Moreover, the longer interatomic distance and other unique microstructural features often help to obtain a lower elastic modulus and outstanding fracture strength [218–222]. Different characterization techniques, such as nanoindentation, the micro-hardness test, and compressive and tensile tests, are performed to extract the properties of metallic-glass systems. The mechanical properties of various Zr-, Ti-, Fe-, Mg-, Cu-, Pd-, Al-, Ni-, Ca-based systems have been listed in Table 7.

**Table 7.** Mechanical properties of various metallic-glass systems. U, C, Y, and F indicate ultimate, compressive, yield and fracture strengths, respectively. For hardness values, V and C indicate Vicker's and compressive hardness, respectively.

| Name of the Systems | | Strength (GPa) | Elastic Modulus (GPa) | Hardness (GPa) | Measurement Technique | Ref. |
|---|---|---|---|---|---|---|
| Zr-based MGs | $Zr_{41.25}Ti_{13.75}Cu_{12.5}Ni_{10}Be_{22.5}$ | 1.9 (U) | 96 | 5.23 (V) | Tensile and compression test | [223] |
| | $(Zr_{55}Al_{10}Ni_5Cu_{30})_{98.5}Si_{1.5}$ | 1.8 (U) | 87 | 5.2 (V) | Tensile and compression test | [224] |
| | $Zr_{61}Cu_{17.5}Ni_{10}Al_{7.5}Si_4$ | 1.8 (C) | -- | 5 (C) | Compressive strength and hardness | [75] |
| | $Zr_{46}Cu_{37.6}Ag_{8.4}Al_8$ | 1.9 (Y) | 92 | 5.4 (V) | Vickers microhardness | [216] |
| | $Zr_{40}Ti_{14}Ni_{10}Cu_{12}Be_{24}$ | 2.3 (Y) | 114 | 9.7 | Nanoindentation | [225] |
| | ZrTiCuNiAl | 1.96 (Y) | -- | 5.5 | Nanoindentation | [226] |
| | $Zr_{52.5}Al_{10}Ti_5Cu_{17.9}Ni_{14.6}$ | 0.82 (Y) | 109 | -- | Nanoindentation | [227] |
| | $Zr_{69.5}Cu_{12}Ni_{11}Al_{7.5}$ | -- | 93.86 | 5.66 | Nanoindentation | [228] |

<div align="center">

**Table 7.** *Cont.*

</div>

|  | | | | | | |
|---|---|---|---|---|---|---|
|  | ZrTiAlFeCuAg (Zr = 60%) | 1.58 ± 0.03 (Y) | 78 ± 1 | 4.5 ± 0.06 (V) | Compression, notch-toughness tests, and ultrasound spectroscopy | [214] |
| Ti-based MGs | $(Ti_{40}Zr_{10}Cu_{38}Pd_{12})100-x Nb_x$ (x = 0, 2, 3, 4) | 1.2–2 (F) | 100–106 | 6–8 (V) | Compression test | [229] |
|  | $Ti_{40}Zr_{25}Ni_3Cu_{12}Be_{20}$ (10nm/2.50mN/s) | -- | -- | 8.29 ± 0.13 | Nanoindentation | [230] |
|  | $Ti_{75}Zr_{10}Si_{15}$ | 2.6 (Y) | -- | 0.007 | Microhardness | [54] |
|  | $Ti_{60}Nb_{15}Zr_{10}Si_{15}$ | 2.2 (Y) | -- | 0.006 | Microhardness | [54] |
|  | $Ti_{40}Cu_{36}Pd_{14}Zr_{10}$ | -- | 110 | 7.7 | Microscratch | [231] |
|  | $Ti_{50}Cu_{25}Ni_{15}Sn_3Be_7$ | 2.17 (C) | -- | 6.5 | Uniaxial compression | [232] |
|  | $Ti_{47}Cu_{38}Zr_{7.5}Fe_{2.5}Sn_2Si_1Ag_2$ | 2.08 (C) | 100.4 ± 0.1 | 5.7 ± 0.05 | Compression | [178] |
| Cu-based MGs | $Cu_{60}Zr_{30}Ti_{10}$ | 1.78 (Y) 2 (F) | 112 | 6.4 (V) | Tensile and compression deformation | [233] |
|  | $Cu_{60}Hf_{25}Ti_{15}$ | 1.92 (Y) 2.13 (F) | 120 | 6.6 (V) | Tensile and compression deformation | [233] |
|  | $Cu_{60}Zr_{30}Ti_{10}$ | -- | 93.88 ± 1.7 | 7.61 ± 0.33 | Nanoindentation | [234] |
|  | $(Cu_{0.6}Hf_{0.25}Ti_{0.15})_{90}Nb_{10}$ | 2.073 (Y) 2.232 (F) | 106 | -- | Compression test | [235] |
|  | $Cu_{47}Ti_{33}Zr_{12}Ni_8Si_1$ | 2.087 (F) | 118.6 | -- | Compression test | [236] |
|  | $(Cu_{50}Zr_{50})_{90}Al_{10}$ | -- | 117.3 | 5.3 (V) 8.7 | Microhardness nanoindentation | [237] |
|  | $Cu_{49}Hf_{42}Al_9$ | 2.408 (Y) 2.620 (F) | 102 | -- | Compression test | [238] |
|  | $(Cu_{0.50}Hf_{0.35}Ti_{0.10}Ag_{0.05})_{97}Ta_3$ | 2.510 (F) | 151.2 | $6.0^4$ | Compression test nanoindentation | [239] |
| Fe-based MG | $Fe_{59}Cr_6Mo_{14}C_{15}B_6$ | 3.8 (Y) 4.4 (F) | 204 | ~11 | Compression tests | [222] |
|  | $Fe_{41}Co_7Cr_{15}Mo_{14}Y_2C_{15}B_6$ | 3.5 (F) | 265 | 12.3 (V) | Compression, bending and hardness tests | [240] |
|  | $Fe_{36}Co_{36}B_{19.2}Si_{4.8}Nb_4$ | >4 (Y) | 201 ± 10 192 ± 0.5 | 14 | Compression test nanoindentation | [241] |
|  | $(Fe_{0.75}B_{0.15}Si_{0.1})_{96}Nb$ | 3.25 (Y) | 175 | 10.4 (V) | Compression Test | [242] |
|  | $[(Fe_{0.8}Co_{0.2})_{0.75}B_{0.2}Si_{0.05}]_{96}Nb_4$ | 4.05 (Y) 4.17 (F) | 205 | 12.01 (V) | Vickers hardness compression test | [243] |
|  | $Fe_{66}Mo_{10}P_{12}C_{10}B_2$ | 2.55 (Y) 3.25 (F) | 176 | 8.83 (V) | Microhardness, compression and resonant ultrasound spectroscopy test | [244] |
|  | $Fe_{0.432}Co_{0.288}B_{0.192}Si_{0.048}Nb_{0.04})_{98}Cr_2$ | ~4.0 (Y) | -- | 11.51–12.51 | Nanoindentation | [245] |
| Pd,Ni,Mg,Ca,Al-Based MGs | $Pd_{40}Ni_{40}P_{20}$ | 1.78 (Y) | 103–108 | -- | Nanoindentation | [246] |
|  | $Ni_{60}Nb_{37}Sn_3$ | 3.7 | -- | 8.83 (V) | Vickers hardness measurements | [247] |
|  | $Mg_{66}Zn_{29}Ca_4Ag_1$ | -- | -- | 2.35 ± 0.03 | Microhardness | [248] |
|  | $Ca_{20}Mg_{20}Zn_{20}Sr_{20}Yb_{20}$ | 0.37 ± 0.025 (F) | 19.4 ± 3.4 | -- | Uniaxial compression test | [249] |
|  | $Al_{85}Y_{10}Ni_5$ | 0.92 | 62.8 | 3.7 | Microhardness and tensile | [250] |
|  | $Mg_{65}Cu_{25}Gd_{10}$ | ~0.8 (C) | -- | ~2.5 | Compression test | [75] |

Disordered-state solids have the limitations of a reduced plasticity at room temperature. Metallic glass systems are reported to deform catastrophically under localized shear bands due to their limited number of active shear bands [69,251–254]. The combination of the strength and ductility of disordered solids has been reported to have improved with the

introduction of nano-crystallites [255,256] or nano-quasicrystals [256,257]. Calin et al. [69] suggested that the inclusion of nano-crystalline may be helpful to improve their ductility. A study conducted by Liu et al. [258] established a relationship between the strength and ductility of CuZr-based metallic-glass composites, as shown in Figure 9a. The study indicated that the combination of high strength and improved ductility can be achieved through percolation by matching the length scales of particle size and inter-particle distance in the amorphous microstructure [258]. The relationship between the tensile strength and ductility among different metallic-glass systems is shown in Figure 9b. Furthermore, presence of specific species in a multicomponent system is another crucial factor to consider in achieving strength and ductility in an alloy. Eckert et al. [259] fabricated a binary metallic glass ($Cu_{50}Zr_{50}$) with nano-crystallites and compared its response with a ternary metallic-glass system ($Cu_{47.5}Zr_{47.5}Al_5$). It was reported that the inclusion of Al in the matrix extended the nucleation of its shear bands [259]. In this study, the multiple homogenous nucleation of shear bands in the ternary metallic glass system ($Cu_{47.5}Zr_{47.5}Al_5$) was found to exhibit a better compressive ductility alongside an increased strength [259]. However, to provide a specific hypothesis on how to achieve strength and ductility in a metallic-glass system requires rigorous study and is currently a research challenge.

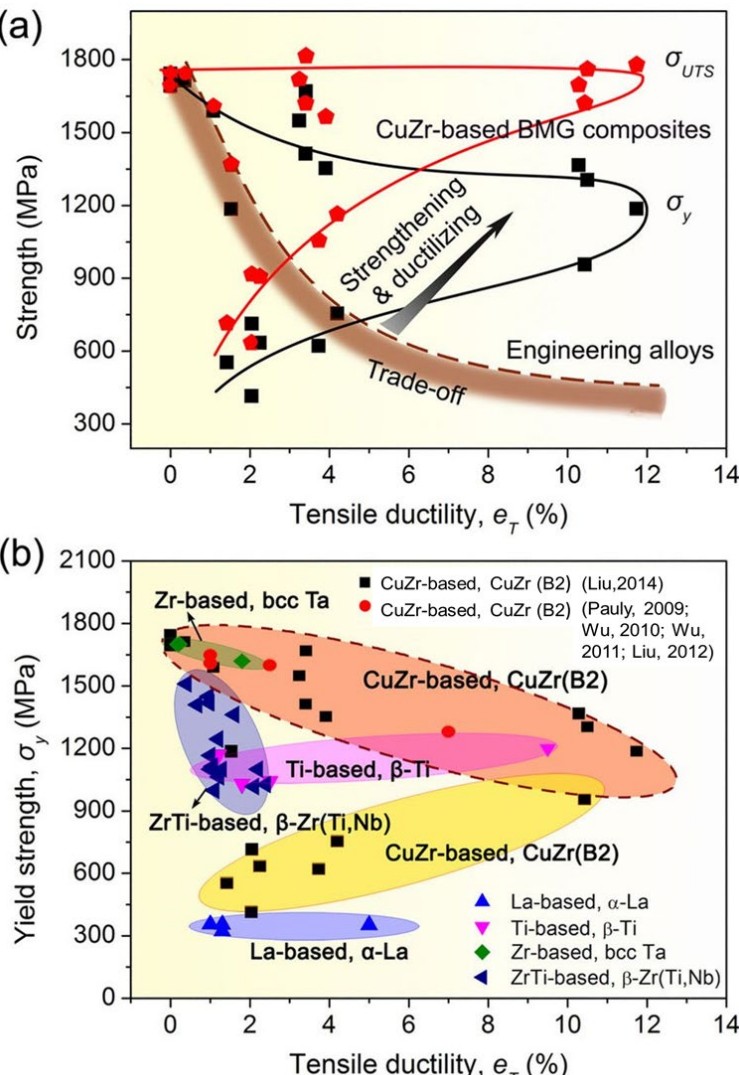

**Figure 9.** (**a**) Strength vs. ductility relationships of Cu-Zr-based metallic glass compared to conventional engineering alloys. (**b**) Tensile properties of different metallic-glass systems. The ductility and yield strength of CuZr—based metallic glass composites is compared with existing literatures [258,260–263].

### 4.4. Electrochemical Properties

Homogeneity and the absence of a long-range order in the microstructure have been found to be advantageous to better electrochemical properties [264,265]. The dissolution rate of a multi-component system depends on the alloying element species, chemical composition, electrolyte chemistry, experimental environment and thermodynamic metastability [35,266–268]. The effects of compositional and chemical homogeneity, as well as the effects of a short- to medium-range order, on the electrochemical responses of disordered solids have been studied by different research groups [29,269]. A study conducted by Kou et al. [270] found a better corrosion resistance in Zr- and Ti-based metallic glasses ($Zr_{41.25}Ti_{13.75}Ni_{10}Cu_{12.5}Be_{22.5}$ and $Ti_{40}Zr_{25}Ni_8Cu_9Be_{18)}$ in NaCl and $H_2SO_4$ solutions compared to a crystalline $Zr_{41.25}Ti_{13.75}Ni_{10}Cu_{12.5}Be_{22.5}$ alloy. It was concluded that chemical homogeneity and the absence of crystalline defects facilitate the passive behavior of metallic-glass systems compared to the crystalline counterpart [270]. Furthermore, metallic glasses were found to be thermodynamically metastable; thus, their structures tend to exhibit the excessive release of free volume [35,271]. A study conducted by Jiang et al. [271] determined the electrochemical behavior of $Zr_{52.5}Cu_{17.9}Ni_{14.6}Al_{10.0}Ti_{5.0}$ metallic glass due to the change in free volume. The study indicated that the reduction in free volume improves the corrosion resistance of metallic-glass systems [271].

Metallic-glass systems were also found to have a higher resistance towards localized or pitting corrosion compared to crystalline materials [220,272]. One of the primary reasons behind pitting corrosion is the presence of physical irregularities in the protective oxide or passive layer [266,273]. Different phases, scratches, grain boundaries, and crystal imperfections can cause damage in the passive layer and eventually trigger catastrophic failure of the systems [274]. The absence of long-range-order and corrosion initiation sites in the lattice structure were found to facilitate the electrochemical properties of metallic-glass systems [275,276].

An assessment of the electrochemical properties of different metallic glasses has been conducted in a simulated environment using different electrolytes for various engineering applications [177,277]. The commonly used electrolytes are phosphate buffer solution (PBS) [266,278,279], Hank's solution [154,229,280], sodium sulfate ($Na_2SO_4$) [58,281], Ringer's solution [275,282], aqueous NaCl solution [209,283], etc. Among the aforementioned electrolytes, PBS (pH = 7.4), Hank's (pH = 7.4) and Ringer's (pH = 6.5) solutions are commonly used to explore the electrochemical behavior of metallic-glass systems for bio-implant applications, whereas 0.05 M $Na_2SO_4$ and 0.6 M NaCl mimic humid air and simulate seawater environments, respectively [154,177,229,278]. Among various glass-forming materials, Zr-, Ti-, and Fe- based metallic-glass systems have been extensively studied [272,284–290]. The enthusiasm behind the selection of these metallic-glass systems has been attributed to their higher glass-forming abilities [20,291,292], lower ion dissolution rates, and stronger and faster oxide-forming capabilities within a wide pH range [293–295]. In addition, the presence of oxide-forming and noble species can also influence the rate of corrosion. The presence of Nb [296], Ag [169], Cu [297], Co [298], Cr [290,299], Ni [300], Mo [277], Be [301], Al [302], W [283], and Pd [229] in metallic-glass systems have been found to improve their electrochemical properties. A study conducted by Zhang et al. [303] explored the effect of Ti, Cr, Mn, Fe, Co, Ni and Cu inclusion in an Al-based metallic-glass system. Its polarization curves (as shown in Figure 10) indicate excellent corrosion resistance, a higher pitting potential, and a comparable or lower corrosion current density in the metallic-glass systems compared to pure Al [303]. The study concluded that engineered metallic-glass systems with quasi- or nano-crystalline phases are electrochemically superior to conventional crystalline materials [303].

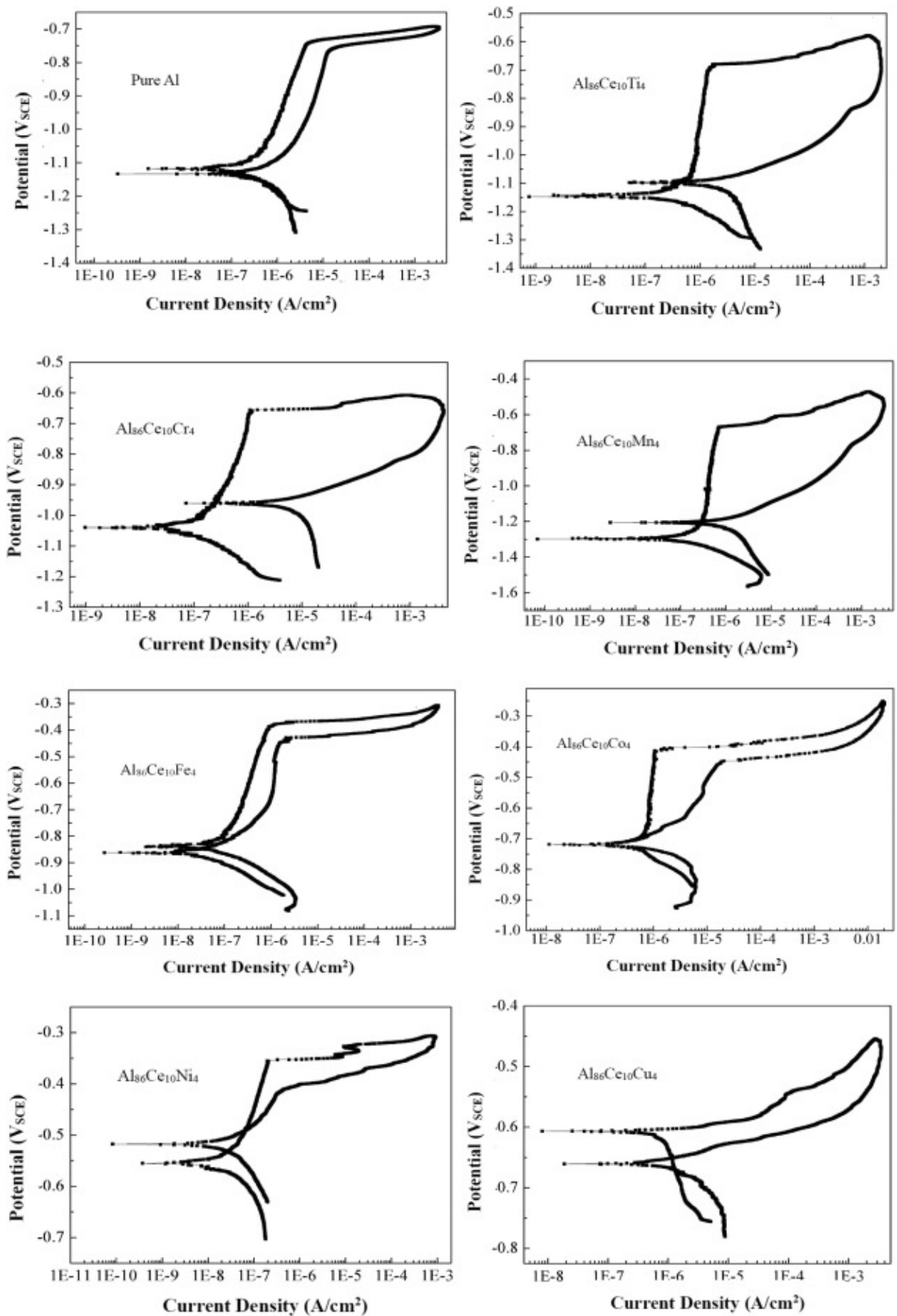

**Figure 10.** Electrochemical analysis of pure aluminum (Al) and different Al-based amorphous systems ($Al_{86}Ce_{10}X_4$, where X = Ti, Cr, Mn, Fe, Co, Ni and Cu) in NaCl (3.56 wt.%) [303].

### 4.5. Magnetic Properties

Excellent soft magnetic properties make metallic-glass systems ideal candidate materials for sensors, actuators, and magneto-optic storage applications [304–306]. The magnetic properties of metallic-glass systems were found to be improved by thermal annealing [307]. The significance of thermal annealing is the formation of nano-crystallites within the amorphous system [307]; thus, the amorphous–crystalline composite microstructure shows enhanced softness compared to a completely disordered microstructure [308]. The size and distribution of the nano-crystals also influence the overall magnetic properties [309]. A mathematical model by Jin et al. [309] explained that the grain orientation changes the coercivity and remanence, and thus, modifies the magnetic properties of a system.

Fe- [310,311] and Co-based [312] metallic-glass systems are commonly investigated due to their outstanding magnetic features, such as their low coercivity, high electrical resistivity, and saturation magnetization [304]. These inherent magnetic properties are also dependent on the chemical composition and atomic configuration of the species of the amorphous systems [129]. A study conducted by Zhu et al. [313] showed a relationship between the coercivity and saturation magnetization of Fe-based metallic-glass systems due to the inclusion of various amounts of Mo (as shown in Figure 11). A higher saturation magnetization is attributed to the modification of the local moment due to the inclusion of Mo. In addition, the elimination of pinning spots creating domain walls was found to be useful in achieving a lower coercivity, which yields soft magnetic properties to the system [313]. Furthermore, Shen et al. [314] explored the magnetic properties of Fe-based Fe–Ga–P–C–B–Si metallic-glass systems in detail. The study indicated that Fe-based metallic glasses exhibit an improved Currie temperature, saturated magnetization, and coercive force [314], which are indicative of good soft magnetic properties. Another study conducted by Lin et al. [315] considered the influence of Yttrium (Y), Dysprosium (Dy), Holmium (Ho), Erbium (Er), and Scandium (Sc) inclusion in an Fe-B-based alloy. The study reported that the species with 130% atomic size mismatched with Fe, and a eutectic point with both Fe and B may impart high saturation magnetization and electrical resistivity, and low coercivity to the amorphous Fe-based solids [315]. Furthermore, a Co-based metallic-glass system (Co–Fe–Ta–B) studied by Inoue et al. [316] exhibited low coercive force (0.25 A/m) and high permeability (550,000). The promising soft magnetic properties of the Co-based metallic glass were attributed to the homogenous atomic structure of the system [317]. The magnetic properties of different Fe- and Co-based metallic-glass systems are listed in Table 8.

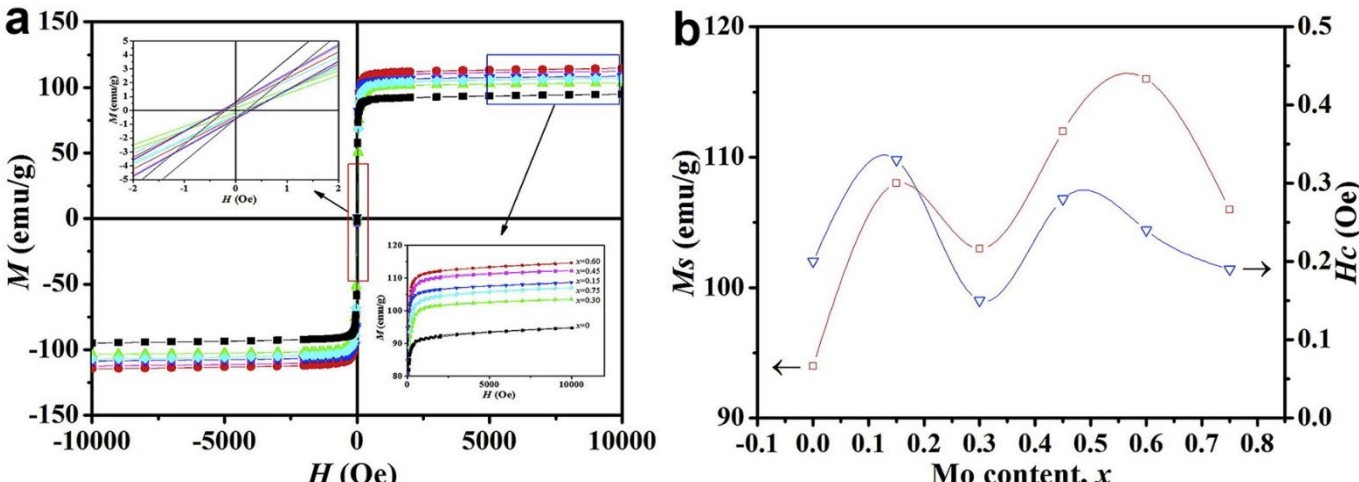

**Figure 11.** Soft magnetic features of Fe-based ($Fe_{80}(Nb_{1-x}Mo_x)_5B_{15}$; where, x = 0, 0.15, 0.30, 0.45, 0.60, 0.75) metallic glass. (**a**) M-H hysteresis loops indicating the relationship between saturation magnetization ($M_s$) and coercivity ($H_c$), and (**b**) the variation in $M_s$ and $H_c$ with the Mo content [313].

**Table 8.** Magnetic properties of various Fe- and Co-based metallic-glass systems.

| | Name of the Systems | Coercivity (A/m) | Saturation Magnetization (T) | Permeability | Currie Temperature (K) | Ref. |
|---|---|---|---|---|---|---|
| Fe-based | $Fe_{70}Al_5Ga_2P_{9.65}C_{5.75}B_{4.6}Si_3$ | 2.2 | 1.2 | 110,000 | 620 | [318] |
| | $[(Fe_{1-x}Co_x)_{75}B_{20}Si_5]_{93}Nb_4Y_3$ Ribbon (x = 0, 0.2, 0.4, 0.6) | 2–15 | ~0.4–0.6 | -- | -- | [319] |
| | $Fe_{72}Al_5Ga_2P_{11}C_6B_4$ | 5.1 | 1.07 | 9000 | 596–605 | [320] |
| | $[(Fe_{1-x}Co_x)_{0.75}B_{0.2}Si_{0.05}]_{96}Nb_4$ | 1.5–2.7 | 0.84–1.13 | $>1.2 \times 10^4$ | 600–690 | [321] |
| | $Fe_{76}Si_9B_{10}P_5$ | 0.8 | 1.51 | -- | -- | [322] |
| | $Fe_{76-x}C_{7.0}$-$Si_{3.3}B_{5.0}P_{8.7}Cu_x$ (x = 0, 0.3, 0.7, 1.0 at.%) | 11 | 1.79 | -- | -- | [323] |
| | $Fe_{66}Co_{10}Mo_{3.5}P_{10}C_4B_4Si_{2.5}$ | 1.0 | 1.23 | 450,000 | -- | [324] |
| | $Fe_{82.75}Si_4B_8P_4Cu_{1.25}$ | 2.1 | 1.83 | 31,600 | -- | [325] |
| | Fe-Si-B-M (M = Cu, Nb, Mo, W, Ta) | 6.9 | 1.41 | 6000 | 631 | [326] |
| | $Fe_{55}Co_{30}Cu_1Nb_7Si_1B_8$ | 5.9 | 1.71 | 1150 | -- | [327] |
| Co-based | $Co_{66}Fe_4Mo_2Si_{16}B_1$ | -- | -- | ~109,000 | -- | [328] |
| | $Co_{bal}Fe_4Ni_2Si_{15}B_{14}$ | <1–2 | -- | 24,000 | 490 | [329] |
| | $Co_{33.9}Fe_{33.9}B_{22.5}Si_{5.7}Nb_4$ | 4.9 | 0.98 | -- | -- | [330] |
| | $Co_{40}Fe_{27}Zr_3Ti_3Mo_{1.5}Si_{1.5}B_{24}$ (Cylinder) | 8 | 1.2 | -- | -- | [331] |
| | $Co_{42}Fe_{20}Hf_3Mo_3Ti_3B_{29}$ | 2 | 0.6 | -- | -- | [332] |
| | $Co_{67}Fe_4Mo_2Si_{17}B_{11}$ (annealed at 360°) | ~0.1 | ~1.2 | -- | -- | [333] |
| | $Co_{68.15}Fe_{4.35}Si_{12.5}B_{15}$ | 210 | 0.81 | -- | -- | [334] |
| | $Co_{43}Fe_{20}Ta_{5.5}B_{31.5}$ | 0.25 | 0.49 | 550,000 | -- | [335] |

## 5. Applications of Metallic Glasses

### 5.1. Biomedical Applications

Metallic glasses can be a perfect alternative to conventional crystalline biomaterials (such as 316L stainless steel, Ti or Ti-based alloys, Zr or Zr-based alloys, Co-Cr alloys, etc.) when used as coatings for surgical devices and implants inside the human body. The following two sections (Sections 5.1.1 and 5.1.2) highlight some applications of metallic glasses in biomedical sectors.

### 5.1.1. Antibacterial Application

Nosocomial infections are often escalated due to bacterial infections from medical instruments or devices. Commonly known biomaterials have been found to be ineffective at preventing bacterial infections [58,336–338], whereas metallic-glass systems exhibit excellent antibacterial properties. This unique characteristic is attributed to the composition of their multicomponent amorphous systems, their lower surface roughness, and the presence of antibacterial species in the matrices [63,168,265]. Liu et al. studied the antibacterial capabilities of Zr-Cu-Al-Ag systems, and the observed antibacterial responses are shown in Figure 12 [168]. The $Zr_{38}Cu_{36}Al_{18}Ag_8$ system, which had a lower surface roughness, was found to have the highest antibacterial activity in this study [168]. Hydrophobic surfaces are well known for their better antibacterial response, and metallic-glass systems have been found to exhibit better wettability compared to crystalline materials due to their disordered microstructure [339]. The presence of certain species, such as silver (Ag) and copper (Cu), in a multicomponent system provides excellent resistance towards bacterial attack [59,169,340]. For example, a $Zr_{39}Cu_{39}Ag_{22}$ metallic glass was found to be very efficient against *S. aureus* [58], whereas $Zr_{61}Al_{7.5}Ni_{10}Cu_{17.5}Si_4$ was found to be efficient against *S. aureus*, *E. coli*, *A. baumannii*, *P. aeruginosa*, and *C. albicans* bacteria [341]. Another study conducted by Chu et al. [63] compared the antibacterial activity of Zr-based ($Zr_{53}Cu_{33}Al_9Ta_5$) and Cu-based ($Cu_{48}Zr_{42}Ti_4Al_6$) metallic-glass systems. The result showed that the adhesions of *E. coli* and *S. aureus* bacteria were greatly hindered on metallic glasses

than those on a bare Si wafer [63]. For *E. coli*, both metallic-glass systems showed a 100% antibacterial activity, whereas the Zr-based metallic glass was more antibacterial towards the *S. aureus* than the *E. Coli* bacteria [63].

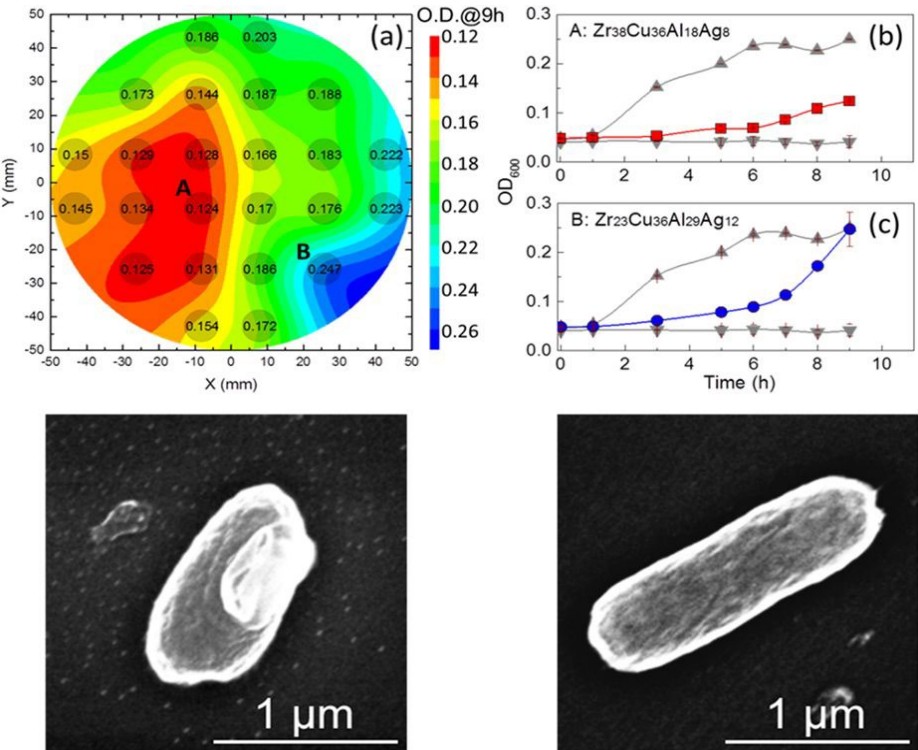

**Figure 12.** Antibacterial activity of Zr−Cu−Al−Ag metallic-glass systems at 9 h. (**a**) Contour plots of optical density (OD) of all the metallic-glass systems. (**b**,**c**) OD vs. time curves representing the highest (red) and lowest (blue) antibacterial activity of the $Zr_{38}Cu_{36}Al_{18}Ag_8$ and $Zr_{23}Cu_{36}Al_{29}Ag_{12}$ systems, respectively, where the controls are shown by the grey lines. The SEM morphologies of E. coli bacteria on the surface of $Zr_{38}Cu_{36}Al_{18}Ag_8$ (**bottom left**) and $Zr_{23}Cu_{36}Al_{29}Ag_{12}$ (**bottom right**) systems, respectively [168].

Commercially used surgical blades are usually made of stainless steel that contains micron-scale roughness on the edge tips and surfaces. The roughness hinders the smooth cut of the soft tissues, and the resulting wear and tear are often difficult to recover. Such limitations of surgical blades and scissors can be resolved by using metallic-glass systems, which exhibit exceptional surface characteristics [179]. A study conducted by Tsai et al. [179] reported lower roughness values, a higher blade sharpness index, and a lower depth of indentation when using a Zr-based metallic-glass system ($Zr_{48}Cu_{35.3}Al_8Ag_8Si_{0.7}$). All of these characteristics make metallic-glass systems an optimum solution for surgical instruments without a compromise in their performance [179].

### 5.1.2. Bio-Implants

Durability and biosafety are two characteristics that are extremely desirable for implantable materials. Elastic modulus mismatch between an implant and a bone, and a lack of resistance towards localized corrosion have hindered the growth of conventional crystalline materials (such as 316L stainless steel, Co-Cr alloys, and Ti-based alloys) as premium choices for implantable materials. The lower corrosion resistance of crystalline materials is often attributed to the presence of grain boundaries and phase precipitates, which act as preferred regions for adverse electrochemical reactions [35,54,336,342,343]. Furthermore, the short interatomic distances of crystalline materials result in higher elastic modulus values, which may lead to failure of the implants due to a stress shielding effect [218]. Metallic-glass systems have the potential to mitigate this problem due to their

homogenous microstructures, with longer interatomic distances and the absence of grain boundaries [35].

Due to their excellent electrochemical and mechanical properties, metallic-glass systems are being used as vascular stents, and dental and orthopedic implants. Zr- [212,273,273,284,344,345], Fe- [275,290,346], and Ti- based [347–349] metallic-glass systems have been widely explored in in vitro conditions for permanent implants. The biological responses of pre-osteoblast cells (MC3T3-E1) [180,350–353], fibroblast cells (L929 and NIH3T3) [302,344,346,354], human-osteoblast-like cells (SaOS2 and MG63) [348,355,356], and endothelial cells [297] reveal the outstanding bio-compatible characteristics of different multicomponent metallic-glass systems. Qiu et al. reported excellent mechanical (improved strength and plasticity), electrochemical (a low passive current density and high pitting potential), and biocompatible responses of a Zr-based system ($Zr_{60}Cu_{22.5}Pd_5Al_{7.5}Nb_5$) in an embryonic-mouse-fibroblast cell line (NIH3T3 cell) [285]. The better biocompatibility of the metallic-glass system was attributed to the short/medium-range order and oxide-forming capability of the amorphous structure [285]. Fe-based metallic-glass systems, studied by Li et al. [290], exhibited better biocompatibility towards NIH3T3 cells and better electrochemical responses in artificial saliva compared to conventional biomaterials. Furthermore, Zr-based metallic glasses have been considered for cardiovascular stents, and studied for endothelial and muscle cells [297]. A cell-morphology and cell-metabolic-activity assessment, as shown in Figure 13, revealed the faster growth of endothelial (HAECs) cells on the Zr-based metallic glasses than on 316 L stainless steel, whereas the growth of smooth muscle (HASCMs) cells was relatively slower [297]. That study reported higher endothelial cell-adhesion capabilities on the Zr-based metallic glasses compared to their conventional crystalline counterpart [297].

Research on biodegradable metallic glass is also gaining a lot of enthusiasm. The biocompatibility, mechanical properties, and electrochemical responses of Mg- [357–361], Ca- [362,363], Sr- [364], and Zn-based [365] degradable alloys have been studied extensively by several research groups. However, researchers investigating biodegradable metallic-glass systems used as fully functional bio-implants still face significant research challenges in their obtaining optimum mechanical and electrochemical properties. For example, bio-implants typically require a higher strength; however, degradation due to pitting corrosion creates surface defects that lead to a gradual loss of strength [35]. In addition, there is a possibility of tissue damage due to hydrogen evolution under a body-fluid environment [35]. Therefore, a proper understanding of this degradation mechanism and the relationship between its strength and degradation rate are required.

## 5.2. Electrochemical Devices

The lower efficiency and durability of catalysts are two primary obstacles that hinder the growth of electrochemical devices to meet the rising energy demand. Owing to their outstanding electrocatalytic activity and durability, metallic glasses can be considered prominent candidates for energy-storage and -conversion devices, such as fuel and electrolysis cells, and batteries. Nevertheless, some key components of electrochemical devices, such as the membrane, catalyst, and separator, still require further development to resolve issues associated with weight and cost [366]. For example, proton-exchange membrane (PEM) fuel cells are suitable candidates for power-generation applications [367–369]. PEM fuel cells consist of bipolar plates that are electrically conductive and essential to isolating the fuel and oxidant gases. Bipolar plates are conventionally made of carbon graphite; however, its brittleness and high production cost limit its use. To overcome this limitation, Kim et al. [367] investigated $Ni_{65}Cr_{15}P_{16}B_4$-metallic-glass-coated plates fabricated using HVOF spray coating and a subsequent hot-pressing technique. Enhanced corrosion resistance and durability suggest the potential of a multicomponent metallic-glass system to be used as bipolar plates in fuel cells [367]. Another prerequisite of a fuel cell is efficiency, and ultra-high-purity hydrogen was found to be crucial to enhancing its efficiency [370]. Crystalline materials are currently being used for hydrogen-storage applications [371].

But the embrittlement of crystalline materials limits its applications. A novel multicomponent metallic glass could be an alternate choice, as the glassy matrix of metallic glass possesses numerous sites for hydrogen absorption [371] and requires no modification to its microstructure [372]. A study conducted by Jayalakshmi et al. [370] explored Ni-Nb metallic-glass systems for hydrogen-related energy applications. The study determined the higher absorption-capacity and embrittlement-resistance characteristics of metallic-glass systems, which induce the better interaction of hydrogen with the metals [370]. Moreover, high permeability and a lower dissolution of metallic glasses make them ideal choices for hydrogen-permeable membranes and separators of fuel cells, respectively [370].

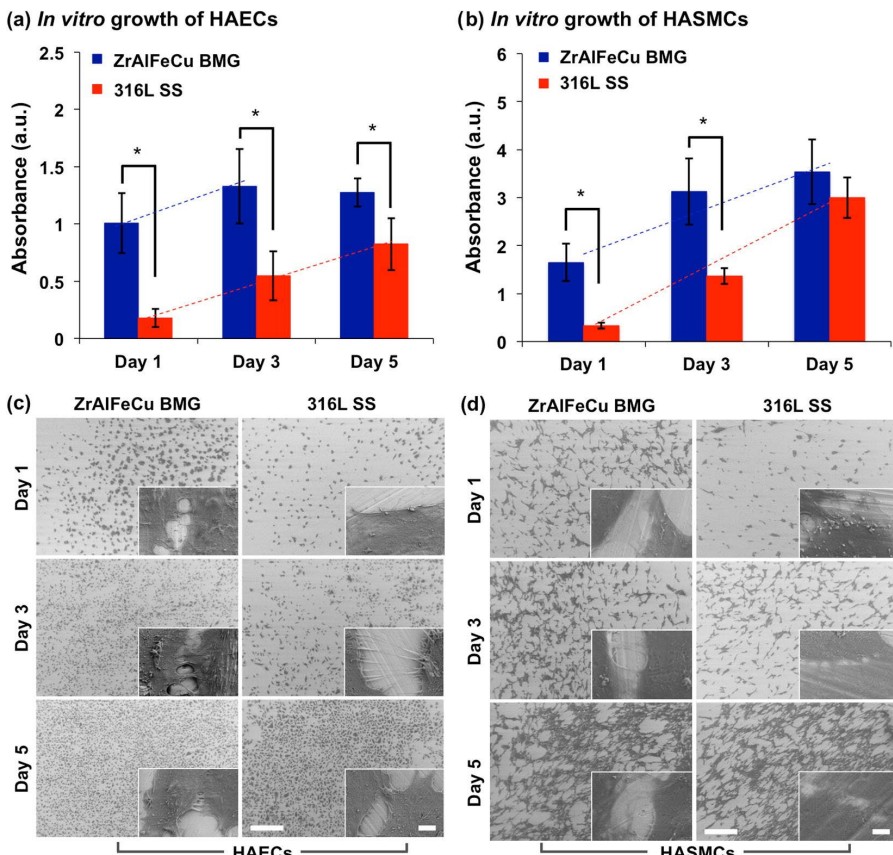

**Figure 13.** Metabolic activity of (**a**) HAECs and (**b**) HASMCs at day 1, 3 and 5, where (*) represents statistically a significant difference ($p < 0.05$; n = 3). SEM images depicting cell morphology at 1, 3, and 5 days for (**c**) HAECs and (**d**) HASCMs. (Scale bars = 500 μm and 5 μm for the larger and inset images, respectively) [297].

*5.3. Optoelectronic Devices*

Metallic-glass systems are gaining substantial momentum in the micro- and nanoimprinting of optoelectronic devices [373]. Smooth surface conditions and the negative enthalpy of mixing of metallic-glass systems facilitate optical transmittance and reflectivity, which are essential for different optoelectronic devices [304]. In the optoelectronic industry, indium-tin-oxide (ITO) is a prominent choice due to its excellent transparency and conductivity [374]. ITO films are also often used in solar cells and collectors, automobile windows, camera lenses, and lamps [374,375]. However, the cost of using ITO films is very high and the use of a metallic glass structure can be a prominent alternative [304]. Huang et al. studied a bi-layer ITO/ZrCu structure, which achieved good conductivity and transparency [376]. The study discovered that the lower resistivity and negative enthalpy between the atoms of the metallic-glass system can be beneficial by forming a continuous layer for transparent conductor design [376]. Wang et al. [377] investigated a $Ag_{40}Mg_{18}Al_{42}$ metallic glass system and found that the lower surface roughness, atomic defects, free

volumes, and electric resistivity improved its optical reflectivity. It is important to note that some of these features, such as surface morphology, atomic structure, and chemical composition, depend on fabrication routes and can be improved further by post-heat-treatment processes [376,378].

Chalcogenide glasses have found numerous applications in optical devices as well. These unique glasses are based on chalcogen elements such as S, Se, and Te, and are formed by the addition of Ge, As, Sb, and Ga and doped by rare-earth elements. A review article by Seddon [379] discusses their fabrication in bulk, fiber, and film form, their optical and thermal properties, and their applications.

### 5.4. Aerospace Application

The use of disordered-state solids in aerospace applications is gaining interest, although no concrete study has been conducted on any multi-component system until recently. Their high strength and lightweight attributes extend the demand of metallic-glass systems into lighter, smaller, and cost-effective aerospace applications [210]. Research conducted by Axinte [210] indicated that the aircraft and spacecraft fasteners can be produced from metallic glass. Another study by Burgess et al. [380] indicated that the higher strength and hardness of metallic-glass systems may be useful for coatings in aerospace applications. Aluminum alloys (Al-6061 and Al-7075) are widely used in automotive and aerospace industries due to their lighter weight and enhanced thermal conductivity [381]. Telford studied the possibility of using Al in a metallic-glass combination [211]. However, its limited glass-forming ability (GFA), higher affinity of oxide formation, and the need for an extreme experimental condition imposed a research challenge on the development of Al-based metallic-glass systems [381]. Vitreloy-1 ($Zr_{41.2}Ti_{13.8}Cu_{12.5}Ni_{10.0}Be_{22.5}$) [11] is the only commercially available metallic glass that is being studied by the US Department of Energy and NASA for aerospace applications [211]. The primary limitation of using metallic-glass systems in industrial applications is their low plasticity, i.e., minimal plastic deformation before catastrophic failure. However, the combination of higher strength and better corrosion resistance may offset this limitation to foster industrial applications.

### 5.5. Memory Storage Devices

Non-volatile memory (NVM) is being used in computers, smartphones, flash memory devices, and other electronic devices. However, the requirement for a higher density and voltage hinders the writing capacity of NVM [382]. To find a suitable alternative, metal-oxide thin films are being used as storage devices or resistive random-access memory [383–385]. Tulu et al. investigated and optimized a thin-film metallic-glass oxide (TFMGO) for resistive switching and as a multicomponent oxide memory device [386]. A 15 nm thick oxide film of $(ZrCuAlNi)O_x$ was fabricated on a Pt/Ti/Si substrate using magnetron sputtering in the presence of oxygen. The RS I-V curve, as shown in Figure 14a shows the current–voltage formation of a Pt/TFMGO/Pt device with a unipolar behavior of switching. Moreover, its endurance features, as shown in Figure 14b, indicate no reduction in resistance up to 1250 switching cycles [386]. The outstanding features are further confirmed by its retention characteristics, as shown in Figure 14c, which exhibit a good resistance ratio and the regeneration of resistive switching without the need of thermal forming [386]. These excellent behaviors are attributed to the amorphosity of the metallic-glass matrix, which has been confirmed through nano-scale characterization. These findings create an opportunity to use thin-sized metallic-glass systems for storage device applications.

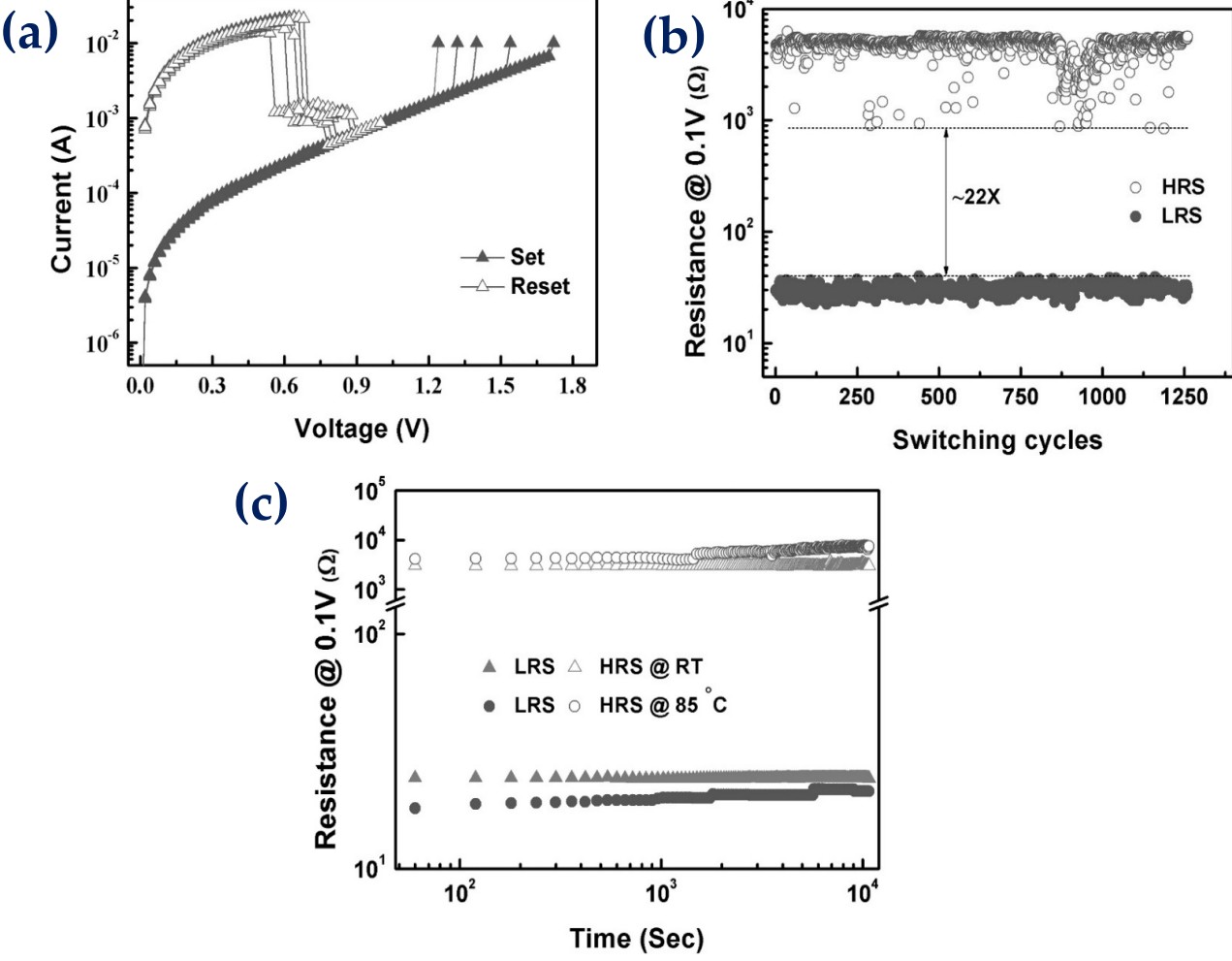

**Figure 14.** (**a**) Unipolar current (I)—voltage (V) curve of a Pt/TFMGO/Pt memory device. (**b**) Endurance and (**c**) retention features measured at a reading voltage of 0.1 V for the Pt/TFMGO/Pt memory cell and for a resistance with respect to the switching cycle and time, respectively [386].

## 6. Conclusions

This article presents a comprehensive review of the history, current state, and prospect of various metallic-glass systems for various engineering applications. The formation of metallic glasses has been discussed in light of the 'confusion principle' of Greer and the empirical rules of Inoue. As postulated by Inoue, three different parameters, i.e., the number of atomic species, atomic size ratio, and negative heat of mixing affect glass formation, and have been discussed in detail; however, metallic-glass formation often deviates from these empirical rules. Different conventional and novel fabrication routes, their prospects, and limitations have also been highlighted. The requirement of a very high critical cooling rate of conventional arc-melting or die casting hinders the commercial production of disordered structures. The issues of the critical cooling rate and limited glass-forming ability can be overcome through combinatorial development. However, the limitation of size restricts the potential industrial applications of metallic-glass systems. The ongoing research is focused on mitigating the size constraints of metallic-glass systems via different welding techniques. Nonetheless, metallic-glass systems have been termed the next-generation materials due to their outstanding mechanical, electrochemical, thermal, magnetic, and optical properties. These properties are attributed to the random atomic arrangements of amorphous solids compared to conventional crystalline materials. Furthermore, the absence of dislocations and grain boundaries in a metallic-glass matrix provides better corrosion resistance in different aggressive mediums. Different features of metallic-glass

systems, their limitations, and their applications are summarized in Figure 15. However, limited plasticity and the possibility of catastrophic failure restrict metallic-glass high-load-bearing structural applications. Therefore, there are ample research opportunities to improve the mechanical properties of amorphous solids. For example, adding minor elements such as; Al and La on the matrix of the multicomponent systems has been observed to enhance the ductility and plasticity of high-entropy alloys [387]. Recently, metallic glass systems by adding La or Ce is fabricated successfully [388], however, the effects of such addition on the physical properties requires further investigation. Similarly, composite structures embedded with nano-particles are also well known for excellent mechanical and microstructural properties [389]. Hence, such use of nano-particles in metallic-glass structures can also be explored. Furthermore, metallic glasses in thin film forms have been gaining popularity due to their unique mechanical and electrochemical properties and make such structures suitable for surgical devices, implants, medical devices, supercapacitors, and battery applications [265,273]. However, the toxicity of certain glass-forming elements poses challenges on their use in biological environments. A higher-saturation magnetization and Currie temperature have also been reported for the metallic-glass systems, which enables them to be used as magnetic materials for different applications. In addition, metallic-glass systems have been found to have wide super-cooled regions with a high glass-transition temperature. These outstanding features make metallic-glass systems distinct alternatives to crystalline materials for different engineering applications, including biological implants, surgical tools, electrochemical, optoelectronic, and memory storage applications.

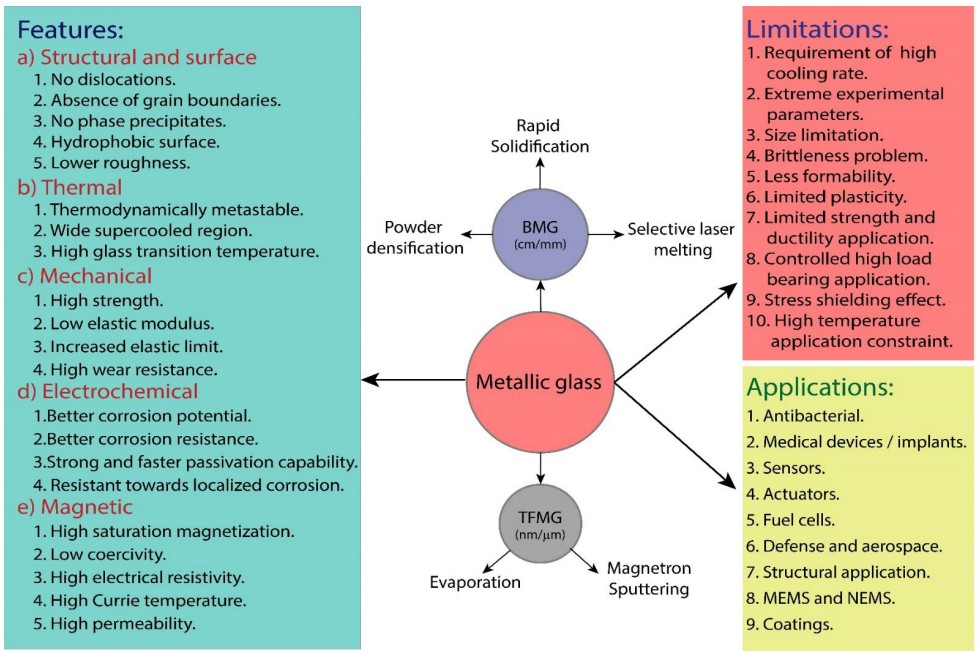

**Figure 15.** Synthesis techniques, features, limitations, and potential applications of metallic-glass systems.

**Author Contributions:** Conceptualization, A.J., I.S. and W.H.; methodology, A.J. and M.N.B.; formal analysis, A.J., M.N.B., W.H. and I.S.; investigation, A.J.; data curation, A.J. and M.N.B.; writing—original draft preparation, A.J.; writing—review and editing, A.J., M.N.B., W.H. and I.S.; visualization, A.J.; supervision, W.H. and I.S. All authors have read and agreed to the published version of the manuscript.

**Funding:** This research received no external funding.

**Institutional Review Board Statement:** Not applicable.



**Informed Consent Statement:** Not applicable.

**Data Availability Statement:** Not applicable.

**Conflicts of Interest:** The authors declare no conflict of interest.

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
