# Peer review of "Distinctive Features and Fabrication Routes of Metallic-Glass Systems Designed for Different Engineering Applications: A Review"

_coatings, doi:10.3390/coatings13101689_

Round 1

Reviewer 1 Report

A review of metallic glasses was made in this work, which summarizes the concepts, fabrications, structure, properties, and applications of metallic glasses. I will sever as a good reference for people interested in this fascinating field of materials science and engineering. I would like to recommend its publication, while the following points are suggested to the authors for consideration:

1) There are many technical issues, leading to "Error! Reference source not found.".

2) Line 40, it is said that the first Au-Si MG was fabricated by Jun et al, while it should be "Duwez et al.".

3) Line 138, "at melting temperature" should be "above the melting/liquidus temperature".

4) Line 158, "solid (alpha-Al) phase" should be "solid solution phase (alpha-Al)".

5) Section 3 is about the fabrication methods of metallic glasses, while section 3.2 is on their welding. I would suggest to put this part at the end of section 3; section 3.3 to 3.5 concentrate more on the fabrication.

6) Line 261 to 264 does not seem to fit with this part, as this paragraph focuses on the surface structures, while reference 145 emphasizes the bulk behavior of MGs.

7) Line 273, the title of section 4.2 could be changed into "thermal stability", since "thermal properties" generally involves thermal conductivity, thermal expansion, and/or thermal capacity. While this section is actually about the thermal stability.

8) For the conclusion part, it would be nice if a summary on the outlook and challenges for future research could be present.

The usage of English is generally acceptable. It would be nice to have it polished further.

Reviewer 2 Report

I have reviewed the manuscript titled " Distinctive features and fabrication routes of metallic glass systems designed for different engineering applications: A review" and I really enjoyed reading this article. I would like to recommend its publication in the journal. Authors in this paper have given an interesting overview of complex alloying systems. Also, they have provided an overview of different applications. The paper is well-written and details are very well discussed. There are however some issues which authors might want to consider to further improve the quality of the manuscript:

1.    Lately, there has been some discussion on the implications of adding minor elements like La and Ce to the structure of metallic glasses. It would be interesting if authors write a few lines on this issue. This paper might be useful for the concept in general: https://doi.org/10.1016/j.jmrt.2023.01.057

2.    Also, there have been some thoughts on using BMG as a matrix for composite materials. It would be great also mention this new development in your review article. This paper might be useful for some backgrounds on the concept in similar systems: https://doi.org/10.1016/j.jclepro.2022.135390

3.    I strongly suggest you add a part at the end of the manuscript before the conclusion, and have some discussions on future developments and research trends (please refer to comment 1 and 2).

4.    I believe the part related to the 3d printing of these alloys needs further attention, as this is a very hot topic in the field.

6.    There is not much papers from MDPI and this journal in your literature, makes me wonder if you have checked MDPI for papers on the topic.

7.    Figure 15 is very nice. Thanks for that.

In the text, I see some typos, like this one “Error! Reference source not found” in line 123. Please have a double check.

Reviewer 3 Report

The article is nicely written. In the introduction please discuss about the other classes of glassy materials like chalcogenide glasses and their advantage and disadvantage.

In the reference section out of several hundred references there was no from (2021-2023). Please add the recent advancement in the filed to make the article comprehensive.

Round 2

Reviewer 2 Report

Thanks for the revision. The revised version is acceptable.